# Full-coverage 250 m monthly aerosol optical depth dataset (2000-2019) emended with environmental covariates by the ensemble machine learning model over the arid and semi-arid areas, NW China

**Xiangyue Chen**[1], **Hongchao Zuo**[1], **Zipeng Zhang**[2], **Xiaoyi Cao**[1], **Jikai Duan**[1], **Chuanmei Zhu**[2], **Zhe Zhang**[2], **Jingzhe Wang**[3]

[1]College of Atmospheric Sciences, Lanzhou University, Lanzhou, 730000 China

[2]Key Laboratory of Oasis Ecology, Xinjiang University, Xinjiang Urumqi 830046, China

[3]School of Artificial Intelligence, Shenzhen Polytechnic, Shenzhen, 518055, China

Correspondence: Hongchao Zuo(zuohch@lzu.edu.cn)and Jingzhe Wang (jzwang@szpt.edu.cn)

## Abstract

Aerosols are complex compounds that greatly affect the global radiation balance and climate system and even human health; in addition, aerosols are currently a large source of uncertainty in the numerical simulation process. The arid and semi-arid areas have fragile ecosystems with abundant dust but lack related high-accuracy aerosol data. To solve these problems, we use the bagging trees ensemble model, based on 1 km aerosol optical depth (AOD) data and multiple environmental covariates, to produce a monthly advanced-performance, full-coverage, and high-resolution (250 m) AOD product (named FEC AOD, Fusing Environmental Covariates AOD) covering the arid and semi-arid areas. Then, based on the FEC AOD products, we analysed the spatiotemporal AOD pattern and further discussed the interpretation of environmental covariates to AOD. The results show that the bagging trees ensemble model has a good performance, with its verification $R^2$ values always remaining at 0.90 and the $R^2$ being 0.79 for FEC AOD compared with AERONET AOD product. The high-AOD areas are located in the Taklimakan Desert and on the Loess Plateau, and the low-AOD areas are concentrated in southern Qinghai Province. The higher the AOD is, the stronger the

interannual variability is. Interestingly, the AOD reflected a dramatic decrease on the Loess Plateau and an evident increase in the south-eastern Taklimakan Desert, while the southern-Qinghai-Province AODs showed almost no significant change between 2000 and 2019. The annual variation characteristics show that the AOD was largest in spring (0.267 ± 0.200) and smallest in autumn (0.147 ± 0.089); the annual AOD variation pattern showed different features, with two peaks in March and August over Gansu Province but only one peak in April in other provinces/autonomous regions. Farmlands and construction lands have high AOD levels compared to other land cover types. Meteorological factors demonstrate the maximum interpretation ability of the AODs on all set temporal scales, followed by the terrain factors, while surface properties have the smallest explanatory abilities; the corresponding average contributions are 77.1%, 59.1%, and 50.4%, respectively. The capability of the environmental covariates to explain the AOD varies seasonally in the following sequence: winter (86.6%) > autumn (80.8%) > spring (79.9%) > summer (72.5%). In this research, we pathbreakingly provide a high-spatial-resolution (250 m) and long-time-series (2000-2019) FEC AOD dataset covering arid and semi-arid regions to support atmospheric and related studies in northwest China; the full dataset is available at https://doi.org/10.5281/zenodo.5727119 (Chen et al., 2021a).

**Keywords:** Aerosol optical depth, Spatial downscaling, Machine learning, Gap filling, Arid areas

# 1 Introduction

Aerosols are a type of complex substance dispersed in the atmosphere that can be from natural or anthropogenic sources (Kaufman et al., 2002). Aerosols can affect the global radiation balance and climate system directly, indirectly, or semi-indirectly by absorbing or scattering solar radiation (Myhre et al., 2013). Concurrently, aerosols seriously endanger human health by mixing, reacting, and dispersing dangerous

compounds (Chen et al., 2020; Lelieveld et al., 2019). As one of the most significant
optical characteristics of aerosols, the aerosol optical depth (AOD) is the integral of the
aerosol extinction coefficient in the vertical direction and indicates the attenuation
impact of aerosols on solar energy (Chen et al., 2021b). The AOD is frequently adopted
to depict air pollution and indirectly calculate various atmospheric parameters, such as
particulate matter 2.5/10; in addition, the AOD is extensively applied in atmospheric
environment-related research (Goldberg et al., 2019; He et al., 2020).
Generally, the primary AOD acquisition method is in-situ observations, which have
high precision. However, in-situ observations are restricted by the distribution of
observation stations, so the resulting data lack spatial continuity, making it difficult to
use these data to meet the objectives of growing regional atmospheric environmental
studies (Zhang et al., 2019). Remote sensing (RS) is an effective tool for collecting
AOD information over a wide range of spatial scales, significantly offsetting the
deficiencies of in-situ observations. RS can tackle difficulties connected to insufficient
data and uneven geographical distributions to a certain extent (Chen et al., 2020).
Nonetheless, RS is not always a silver bullet for acquiring AOD, as RS methods have
some problems, such as low spatial resolutions and missing data in some particular
situations (Li et al., 2020). The commonly utilized AOD satellite products derived from
various sensors have different emphases in their uses (Table S1). However, the common
point is that the spatial resolution of these data is coarse, and the products even contain
large numbers of nodata values (Chen et al., 2022; Sun et al., 2021; Chen et al., 2021b;
Wei et al., 2021). All these issues restrict the application of satellite AOD products on
regional scales and especially on the local scale. Furthermore, the AOD spatial
resolution scale often inevitably affects subsequent atmospheric pollutant predictions
(Yang and Hu, 2018). These issues not only affect AOD analyses but also mislead
numerous pertinent uses of AOD data.
Although methods for resolving AOD RS data deficiencies have been studied,
previous research has not addressed this problem completely (Li et al., 2020; Zhao et
al., 2019). Considerable related work has concentrated on multisource AOD dataset-
fusion or AOD gap-filling methods using different models. The initial and most
extensively applied method is interpolation, but the AODs obtained in this way show
high spatiotemporal variability; thus, this method is not suitable for application to
anticipate missing AOD data (Singh et al., 2017). Another widely used method involves
merging multiple AOD products; this method can improve the data quality but often
fails to completely eliminate missing pixel values, even bringing about offsetting
consequences (Bilal et al., 2017; Ali and Assiri, 2019; Wei et al., 2021). Some statistical
models, such as linear regression and additive models, have also been employed to fill
missing pixel values and improve the spatial resolutions of AOD products. However,
the performances of these models are often dubious due to their simple structures (Xiao
et al., 2017). Most current methods for obtaining high-resolution AOD forecasts are
focused on individual model techniques and rely on a set of assumptions that are
frequently not met, leading to inaccurate predictions (Li et al., 2017; Zhang et al. 2018).
As computing technology advances, involving the training of multiple models by
resampling the training data with the corresponding environmental covariates from
their original distribution, ensemble machine learning methods provide new
considerations and methods that are less constrained by the hypotheses of single models,
with less overfitting and fewer outliers (Li et al., 2018). The strong data-mining ability
of ensemble machine learning methods is also good for fitting multisource data, and
these methods can achieve higher-precision results at the same time (Zhao et al., 2019).
As a result, the present research attempts to adopt ensemble machine learning methods
to explore the production of an advanced-performance, high-resolution, full-coverage
AOD dataset covering arid and semi-arid areas.
Currently, many previous studies have focused on AOD research in various regions
and on various scales, but these studies were concentrated on the eastern coastal areas
and lacked related exploration in arid and semi-arid areas. Arid and semi-arid areas, as
important components of the Earth's geographic units, have extremely fragile

biosystems and are extremely sensitive to climate change and human activities (Huang et al., 2017). Due to the complex surface situation in arid and semi-arid areas, especially those with large desert areas, many AOD retrieval algorithms are not suitable for use in such regions. Although a minority of algorithms can acquire AODs in arid and semi-arid areas, such as the deep blue (DB) algorithm and multiangle implementation of atmospheric correction (MAIAC) algorithm, these algorithms are still limited by their coarse resolution, high uncertainty, or extensive missing-data phenomenon, so the resulting AOD products have difficulty meeting the needs of arid and semi-arid atmospheric environmental research (Wei et al., 2021). However, arid and semi-arid areas are crucial dust sources, with strong variability in the aspects of aerosol loading and optical characteristics. As typical dust sources and AOD data-scarce areas, the AOD variety in arid and semi-arid areas significantly influences global climate change and model simulations. Therefore, manufacturing an AOD dataset covering arid and semi-arid areas with increased quality is necessary for performing local and even global atmospheric environment research.

To better solve the issue associated with the lack of AOD data in arid and semi-arid areas, this research aims to acquire an advanced-performance, high-resolution, full-coverage AOD dataset that can serve as the foundation for future studies. To achieve this goal, the main work of this study includes the following steps: (1) based on the MAIAC AOD product combined with multiple environmental covariates and utilizing a machine learning method, the FEC AODs are obtained for the 2000–2019 period; (2) Aerosol Robotic Network (AERONET) ground observation data and the MCD19A2 and MxD04L2 AOD satellite products are collected to verify the applicability of the FEC AOD product; (3) the FEC AOD spatiotemporal patterns are analysed; and (4) the dominant environmental covariates of the FEC AOD dataset are explored.

## 2 Materials and methods

*2.1 Study area*

Figure 1 shows the arid and semi-arid areas in northwest China (E 73°25' - 110°55',
N 31°35' - 49°15'), a typical arid and semi-arid region on the globe, in terms of their
spatial locations, surface covers, and environmental problems (Ge et al., 2016). As dust
sources and fragile-ecosystem areas, the regional climate differences in this region are
significant, with perennial drought and low-precipitation (< 400 mm) conditions (Ding
and Xingming, 2021). Furthermore, the area is extremely sensitive to climate change
and human activities and has a large AOD variability, which makes global climate
simulations and radiation balance quantifications difficult. With the development of
society and technology, the forces by which people change nature are increasing.
Increasingly unreasonable human activities (such as deforestation and soil salinization)
and poor land management policies (such as reclamation and water resource utilization)
bring about regional vegetation degradation, desertification, rapid glacier melting, and
frequent dust weather, which eventually lead to rapid deterioration of the ecological
environments in all arid and semi-arid areas.

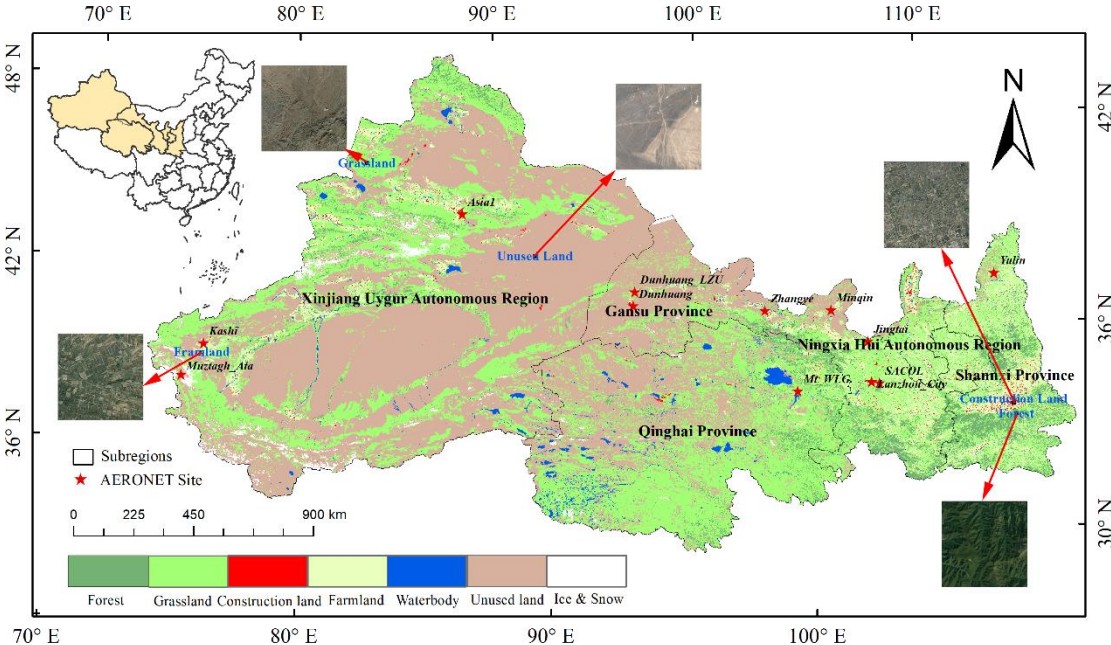


Figure 1. Study area. The figure shows the typical arid and semi-arid areas and
AERONET site distributions, five provinces/autonomous regions in northwest China.
The five ecological zones were derived from Google Earth (https://earth.google.com/).

## 2.2 MODIS MAIAC data

The MAIAC AOD product, which is named MCD19A2, is based on the MODIS instrument onboard Terra and Aqua in combination with the MAIAC algorithm. The MAIAC algorithm is an advanced AOD retrieval method that uses time-series analyses and image-based spatial processing to acquire AOD data from densely vegetated areas as well as bright desert regions (Lyapustin et al., 2018; Lyapustin et al., 2011). The MAIAC AOD product's temporal and spatial resolutions are 1 day and 1 km × 1 km, respectively; this is the highest spatial resolution among existing AOD products. The MAIAC AOD product also offers a long-time-series AOD collection, which is intended for air quality research on regional and even global scales. Compared with former AOD products, the performance of the MAIAC AOD data product on bright surfaces and heavy AOD loading areas is generally considered to reflect significant improvements (Li et al., 2018; Chen et al., 2021b). In this paper, we acquired the MAIAC AOD product for the entire study region from the NASA website (https://search.earthdata.nasa.gov/) over 20 years, from March 2000 to February 2020. Using the Python tool, we pre-processed these data and computed the daily average AOD by combining the 550 nm AOD data from Terra and Aqua.

## 2.3 MODIS MxD04L2 data

MYD04L2 and MOD04L2 are the level-2 atmospheric aerosol products from Aqua and Terra, respectively, and the spatial and temporal resolutions of these products are 10 km × 10 km and 1 day, respectively (Zhao et al., 2021). The MxD04L2 AOD product is constructed using two main algorithms, the Dark Target (DT) and Deep Blue (DB) algorithms, to retrieve the global AOD distribution. Based on MODIS Collection 6.1 data, we chose the 550 nm combined DT and DB AOD to validate the FEC AOD. Notably, the Aqua and Terra launch times are different, so we acquired MOD04L2 data from March 2000 to February 2020, but for MYD04L2, we acquired data only from

July 2002 to February 2020. All processes were realized after downloading data from
the NOAA website (https://ladsweb.modaps.eosdis.nasa.gov/); the calculation and
analysis steps were performed on local computers, and the main works, including the
geometric correction, projection conversion, image mosaicking, clipping, daily and
monthly AOD means computing, and numerical extraction steps, were performed in the
MODIS Reprojection Tool (MRT), ENVI, and ArcGis software.
*2.4 AERONET data*
AERONET (Aerosol Robotic Network) is a network that monitors aerosols on the
ground, providing 0.340-1.060 m aerosol optical characteristics at a high temporal
resolution (15 min) (Holben et al., 1998). AERONET currently includes more than 500
sites and covers the major regions of the world with a long time series. The AERONET
AOD data have a low uncertainty (0.01–0.02) and are considered the highest-accuracy
AOD data available; these data are widely used as a reference in RS AOD product
validations (Almazroui, 2019). In this study, data from a total of 12 AERONET sites in
northwest China were selected, most of which were from the third version of the Level
2.0 AERONET AOD, except for the Mt_WLG station data (Level 1.5) (Yan et al., 2022;
Giles et al., 2019). Related information about these AERONET sites is available in
Table S2 and Figure 1. Satellite products provide mainly 550 nm wavelength AODs, so
the AERONET AOD at 550 nm was computed via the Ångström exponent algorithm to
better match the AODs observed by the satellites (Ångström, 1964). In the temporal
dimension, we computed the average AERONET AODs over the Aqua and Terra
overpass periods. In the spatial dimension, we matched the satellite and in-situ observed
AODs over a 3 × 3 pixel spatial window (Tao et al., 2017). The AERONET data and
related information can be found at https://aeronet.gsfc.nasa.gov.
*2.5 Environmental covariates*
The environmental covariates selected in this study comprised 12 covariates in
three categories (meteorological parameters, surface properties, and terrain factors).
The covariates were selected based on two criteria: first, each variable had to be
considered important to the AOD and to have a vital influence on the AOD formation,
accumulation, and migration processes, referring to existing research and expert
experience (Zhao et al., 2019; Chen et al., 2020; Yan et al., 2022); and second, the data
must be freely released to the public, meaning the datasets must be freely available on
the national or global scale (Li et al., 2020). Detailed information on these covariates is
listed in Table 1. In this study, we computed environmental variable datasets at two
spatial resolutions (1 km and 250 m). The 1 km spatial-resolution data were obtained
with the aim of modelling with the MAIAC 1 km AOD, and the 250 m spatial-resolution
data were the target resolution of the FEC AODs. To normalize the covariables on this
basis, we interpolated the geo-datasets to 1 km and 250 m spatial resolutions in ArcGIS
(the bilinear method was used for the continuous covariates, and the nearest neighbor
method was used for classified covariates) and reprojected the results to the 1984 World
Geodetic System (WGS) coordinates. The environmental covariates were divided into
static and dynamic variables. Static variables were defined as those that did not change
essentially with time, i.e., slowly changing factors. For dynamic covariates, the
averaging method was adopted to obtain monthly average data. The static variables,
similar to the baseline conditions, played an initial constraint role in the downscaling
of the monthly AODs, while the dynamic variables played a more dynamic evolution
role (Yan et al., 2022). Notably, the relevant operations are not limited to ArcGIS, and
relevant open-source software such as QGIS could also be implemented.
**2.5.1 Meteorological parameters**
The meteorological parameters included temperature, precipitation,
evapotranspiration, and wind speed. The temperature and precipitation data were
obtained from National Tibetan Plateau/Third Pole Environment Data Center (TPDC)
at temporal and spatial resolutions of 1 month and 1 km × 1 km, respectively. The
evapotranspiration (ET) data were obtained from the TPDC's terrestrial
evapotranspiration dataset across China at temporal and spatial resolutions of 1 month
and 0.1° × 0.1°, respectively (Szilagyi et al., 2019). For the ET data, we used a
downscaling algorithm proposed by Ma (2017) to transform the values into a 1 km
resolution. The wind speed data were obtained from National Earth System Science
Data Center at temporal and spatial resolutions of 1 month and 1 km × 1 km,
respectively (Sun et al., 2015). For the four meteorological parameters, we calculated
the monthly average value each year for the subsequent research.
**2.5.2 Surface properties**
To describe the surface properties, we employed the land use and land cover
(LUCC), normalized difference vegetation index (NDVI), and temperature vegetation
dryness index (TVDI). From the LUCC dataset, we selected the median year of the
whole study period, 2010, from Resource and Environment Science and Data Center.
The LUCC dataset was obtained by manual visual interpretations of the Landsat-series
data as the data source. This dataset included 6 categories (farmland, forest, grassland,
waterbody, construction land, and unused land) and 25 subcategories at a spatial
resolution of 30 m. LUCC data are often likely to indicate the intensity of human
activity and are closely related to aerosol emissions, transport, and dustfall (Fan et al.,
2020; Li et al., 2022). The NDVI data were obtained from the NASA Global Inventory,
Monitoring, and Modelling Studies (GIMMS) NDVI3g v1 product at temporal and
spatial resolutions of 15 days and 0.083° × 0.083°, respectively. The NDVI data were
downscaled to 1 km, similar to the ET data. The TVDI data were obtained through a
soil moisture inversion method based on the NDVI and surface temperature. This index
can optimally monitor drought conditions and is used to study the spatial variation
characteristics of the drought degree. The temporal and spatial resolutions of the TVDI
data are 1 month and 1 km × 1 km, respectively.

### 2.5.3 Terrain factors

The elevation data were collected from the Shuttle Radar Topography Mission 90 m Digital Elevation Model (SRTM). DEM is highly correlated with surface pressure and always used to represent the dispersion condition of aerosols (Xue et al., 2021; Fan et al., 2020). Based on elevation, geomorphology is realized under Geographic Resource Analysis Support System extension named r.geomorphon modular (Jasiewicz and Stepinski, 2013). Using the System for Automated Geoscientific Analyses software (https://sourceforge.net/projects/saga-gis/), the plan curvature, slope length and slope steepness, and topographic wetness index were computed.


Table 1. Environmental covariates for AOD modelling

| Type | Name | Abbreviation | Resolution | Source |
|---|---|---|---|---|
| **Dynamic covariates** | | | | |
| Meteorological parameters | Temperature | Tem | 1 km × 1 km | http://data.tpdc.ac.cn/ |
| | Precipitation | Pre | 1 km × 1 km | http://data.tpdc.ac.cn/ |
| | Wind speed | WS | 1 km × 1 km | http://www.geodata.cn/ |
| | Evapotranspiration | ET | 0.1° × 0.1° | http://data.tpdc.ac.cn/ |
| Surface properties | Normalized difference vegetation index | NDVI | 0.083° × 0.083° | https://ecocast.arc.nasa.gov/data/pub/ |
| | Temperature vegetation dryness index | TVDI | 1 km × 1 km | http://www.geodata.cn/ |
| **Static covariates** | | | | |
| Surface properties | Land use and land cover | LUCC | 30 m × 30 m | http://www.resdc.cn/ |
| Terrain factors | Elevation | Elev | 90 m × 90 m | http://srtm.csi.cgiar.org/srtm data/ |
| | Geomorphology | Geoms | 90 m × 90 m | |
| | Plan curvature | Curpln | 90 m × 90 m | |
| | Slope length and slope steepness | LS | 90 m × 90 m | |
| | Topographic wetness index | TWI | 90 m × 90 m | |


## *2.6 Bagging tree ensemble*


Ensemble machine learning methods can be divided into two main categories
according to whether dependency relations exist between learners: boosting and
bagging (Figure S1) (González et al., 2020). If there is a strong dependency between
individual weak learners, and a series of individual weak learners needs to be generated
serially (this means that the following weak learner is affected by the former weak
learner), which is boosting. In contrast, if there is no dependency between individual
weak learners, a series of individual learners can be generated in parallel (there is no
constraint relationship between each learner), which is bagging. The typical
representative and extensively used boosting and bagging algorithms are Gradient
Boosting Decision Tree (GBDT) and Random Forest (RF), respectively (Zounemat-
Kermani et al., 2021). Compared to boosting, bagging reduces the training difficulty
and has a strong generalization ability.
Bagging (namely, bootstrap aggregating), as a simple but powerful ensemble
algorithm to obtain an aggregated predictor, is more accurate than any single model
(Breiman, 1996). Bagging uses multiple base learners or individual learners (such as
decision trees, neural networks, and other basic learning algorithms) to construct a
robust learner under certain combined strategies (Li et al., 2018). Generally, bagging
algorithms include bootstrap resampling, decision tree growing, and out-of-bag error
estimation steps. The main steps of bagging are as follows: (1) Bootstrap resampling of
a random sample (return sampling) under abundant individual weak learners; (2) model
training based on the origin samples to train for abundant individual weak learners in
accordance with the self-serving sample set; and (3) outputting the results based on the
decision tree and calculating the average of all the regression results to obtain the final
regression results. Therefore, bagging reduces the overfitting problem and prediction
errors in decision trees and the variance, thereby significantly improving the accuracy
of the results. Simultaneously, the influence of noise on the bagging algorithm is
comparatively lower than those of other available machine learning algorithms for
obtaining AODs (Liang et al., 2021).

300        In this study, we used 12 environmental covariates (1 km) as the downscaling

method (bagging tree ensemble algorithms) inputs to acquire an AOD-environmental
covariate (AODe) model at a 1 km resolution and utilized the AODe model and 250 m
environmental covariates to acquire the FEC AOD product. Specifically, the basic idea
for downscaling AODs with bagging trees ensemble machine learning (ML) models is
to train the relationships between the MAIAC AODs and the auxiliary environmental
variables at a coarse resolution (1 km) using ML algorithms. We then applied the
trained relationships to generate a high-resolution FEC AOD product at a fine
resolution (250 m) (Duveiller et al., 2020; Yang et al., 2020; Ma et al., 2017). In the
case of lacking environmental covariates in some periods, we used the multiyear
monthly average to replace the missing values. The reason why the 250 m target
resolution was selected was that existing studies have shown that the 250-500 m spatial-
resolution scales are appropriate for aerosol RS research and can optimally capture
aerosol features (Wang et al., 2021; Chen et al., 2020). Second, most high-resolution
global product data have a 250 m resolution, especially soil data, as this resolution is
most convenient for peer comparison and further research and application (De Sousa et
al., 2020; Hengl et al., 2017). The model was built monthly from March 2000 to
February 2020 to assure the model's accuracy in the inference process, and the specific
parameter set included 10 cross-validation folds, the number of learners ($N = 30$), and
the minimum leaf size ($L_{min} = 8$). Each base learner was developed using a bootstrap
sample generated individually from the input data. All steps were implemented in
MATLAB R2021a (Figure 2). All modelling and application processes could also be
implemented in R or Python.

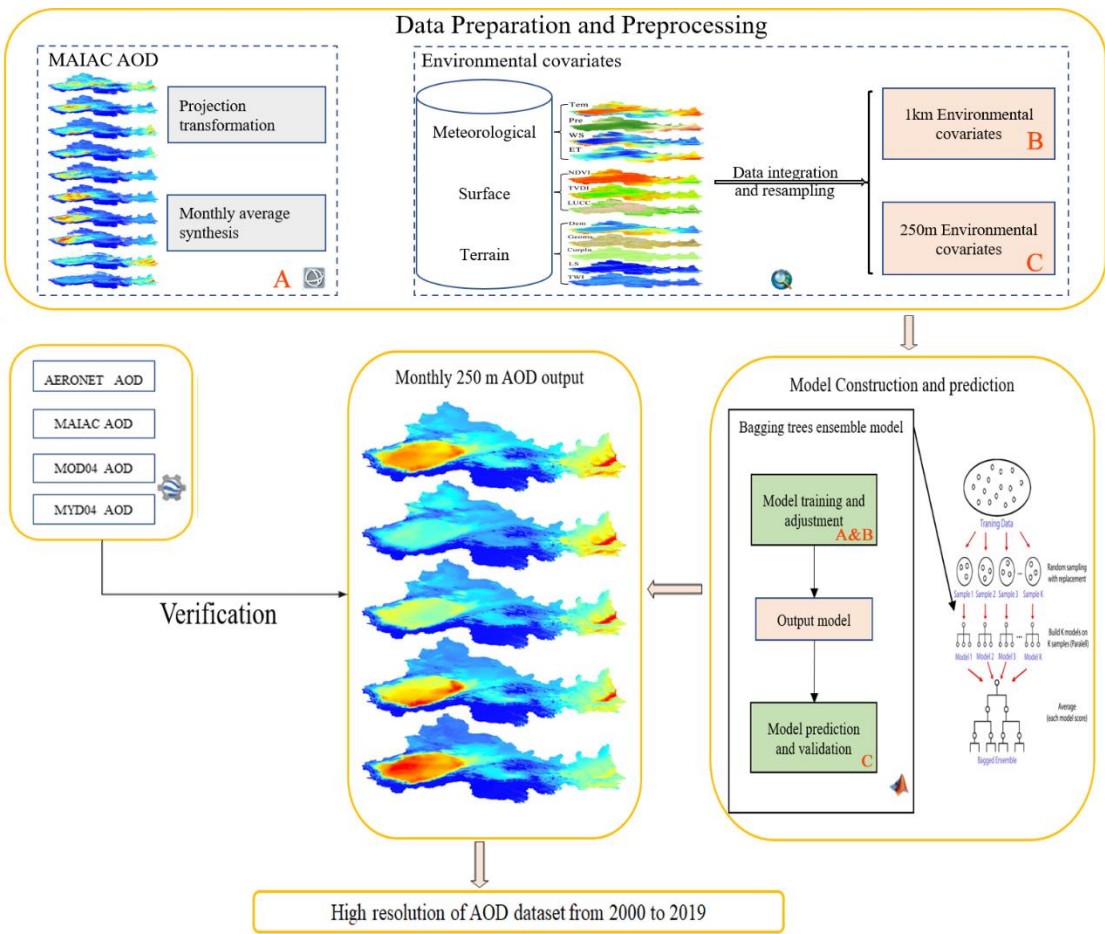


Figure 2. Flow chart of the experiment and model calculation process.

## 3 Results and analysis

### *3.1 Performance evaluation based on in-situ observations*

To verify the performance of the FEC AOD product over arid and semi-arid areas, based on the AERONET AOD data as a reference, some generalized parameters were chosen to assess the performance of the FEC AOD product, such as the decision coefficient ($R^2$), root mean square error (RMSE), and expected error (EE), etc. (Levy et al., 2010; Ali et al., 2019; Feng et al., 2021). When $R^2$ is higher and RMSE is lower, the performance of the FEC AOD is better. The EE can evaluate the degree of overestimation or underestimation of the FEC AOD product via three situations (within EE, above EE, and below EE). To examine the high-resolution and full-coverage FEC

AOD performance, we computed the monthly average AOD at each AERONET site in the whole study region. Specifically, we checked the data time range and data usability at every site, and for the daily scale, we computed the average AOD only from 9:00 am to 2:00 pm local time to obtain the daily mean (if the valid data number in a day was less than 18, the daily mean was considered to be missing). For the monthly scale, if the number of effective daily data was less than 20 days, the monthly mean was considered missing, so 180 effective matching samples were obtained. As shown in Figure 3a, the FEC AODs were highly correlated with the AERONET AODs ($R^2 = 0.787$), with an MAE of 0.049 and RMSE of 0.061. Approximately 83.9% of the monthly collections fell within the EE, with an RMB of 1.018 and a Bias of 0.005, meaning that the FEC AOD product almost overcame some of the overestimation and underestimation problems. Concurrently, we also conducted comparisons between the MAIAC AOD (Figure 3b), MOD04 L2 (Figure 3c), and MYD04 L2 (Figure 3d) products with the AERONET AODs for the same period. The MAIAC AOD product was superior to the MxD04L2 AOD product, and the FEC AODs exhibited obvious improvements compared to the MAIAC AODs, within EE values ranging from 65.0% to 83.9%. It is clear that the performance of the FEC AOD product obviously outperformed the other AOD products in terms of the number of valid data, consistency, and deviation. In addition, compared to previous studies, the FEC AODs also have an improved applicability advantage (Chen et al., 2021b; Wei et al., 2019).

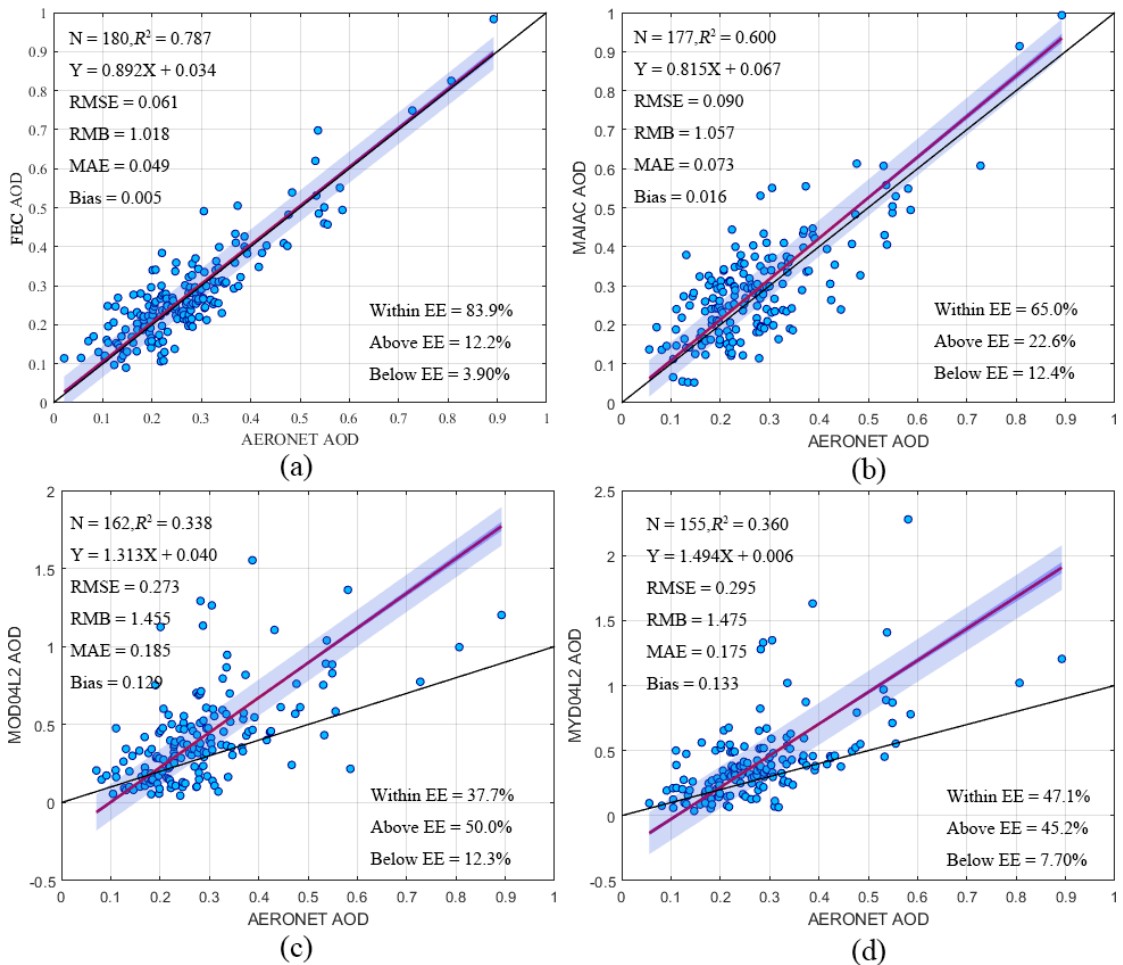

Figure 3. Comparisons of various products with the AERONET AODs: (a) FEC AODs, (b) MAIAC AODs, (c) MOD04L2 AODs, and (d) MYD04L2 AODs. The red line denotes the regression line, the black line shows the 1:1 line, and the blue area indicates the 95% prediction interval.

## 3.2 Comparisons with satellite AOD products

The multiyear average spatial distributions of the FEC AODs, MAIAC AODs, MOD04L2 AODs, and MYD04L2 AODs were calculated (Figure 4). The AOD spatial patterns exhibited high consistency among these products: high AODs were located in the Taklimakan Desert and on the Loess Plateau, and low AODs were distributed in high-elevation areas (such as mountainous zones and Qinghai Province). To further validate the FEC AOD performance, we calculated the monthly, seasonal, and yearly average AODs from 2000 to 2019 (Figure S2-S5). In terms of the monthly scale (Figure S2), we found that many high AOD values appeared in March, April and May, concentrated in and downwind of the Taklimakan Desert. Generally, the FEC AODs,

MAIAC AODs, MOD04L2 AODs, and MYD04L2 AODs had similar monthly spatial
distributions, especially the FEC AODs and MAIAC AODs. The monthly correlations
between the FEC and MAIAC AODs were all above 0.78 in the study area, most of
which were higher than 0.9 ($R_{mean}$ = 0.928, N = 240, P < 0.001) (Figure S3). A similar
spatial pattern also appeared in the multiyear seasonal average AODs (Figure S4).
Spring had the broadest high-AOD-value distribution, followed by summer, while
autumn and winter had high AOD values concentrated on the Loess Plateau. At the
same time, the multiyear annual average AOD also exhibit strong similarity in its spatial
patterns among products (Figure S5). Therefore, we can robustly conclude that the FEC
AOD product has a strong consistency with the MAIAC AOD, MOD04L2 AOD, and
MYD04L2 AOD products with regards to the monthly, seasonal, and yearly average
AOD spatial patterns.

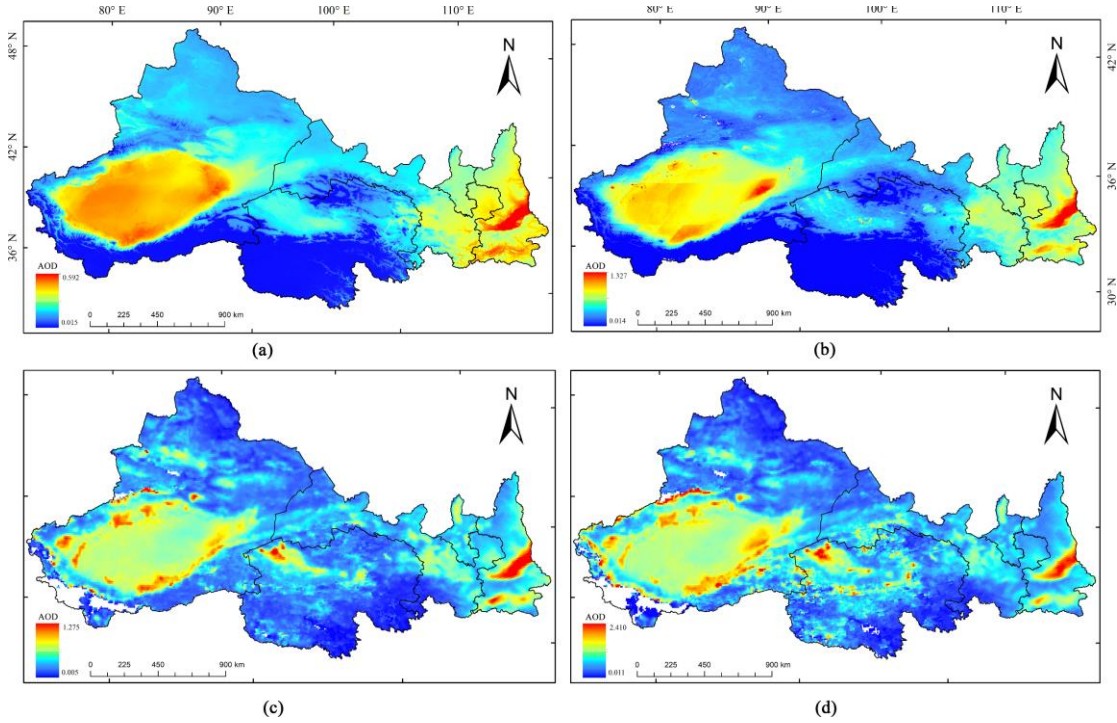


Figure 4. The multiyear spatial average AODs from 2000 to 2019: (a) FEC AODs, (b)
MAIAC AODs, (c) MOD04L2 AODs, and (d) MYD04L2 AODs.

386        Considering that the ability to capture long-term trends is an important element for

a dataset, we compared the FEC AOD, MAIAC AOD, MOD04L2 AOD, and
MYD04L2 AOD products to further validate the FEC AODs. From January to
December, the multiyear monthly averages of these four AOD products showed similar
change trends, increasing and decreasing alternately and reaching their lowest values
in November (Figure S6). Of course, we observed some differences in the AOD
magnitude and fluctuation range; these differences were due mainly to the difference in
AOD retrieval algorithms. To further analyse the consistency and differences among
the products, we also compared the four products on monthly scale by removing the
seasonal cycles (Figure 5). First, the four AOD data products changed in a highly
similar manner, and the MxD04L2 AOD fluctuation range was significantly higher than
those of the FEC AOD and MAIAC AOD products. Notably, the FEC AOD and
MAIAC AOD products were substantially consistent, with an $R^2$ value of 0.953. In
addition, we also computed the monthly and seasonal change trends by removing the
seasonal cycles on the pixel scale. Because MxD04L2 contains a large amount of
missing data and the detrend results show no data, we mainly discuss the spatial change
trends of the FEC AOD and MAIAC AOD products on the monthly and seasonal scales
in the following text. From Figures S7-S8, we find that the monthly and seasonal
change trends reveal good consistency between the FEC AOD and MAIAC AOD
products. The long-term trends based on the monthly AOD data between 2000 and 2019
show a similar spatial pattern in the effective pixels of the FEC AOD and MAIAC AOD
products (Figure 6), with a significant decrease on the Loess Plateau and a significant
increase in the south-eastern Taklimakan Desert. Moreover, the long-term FEC AOD
trends are significant (P < 0.05) in most areas.

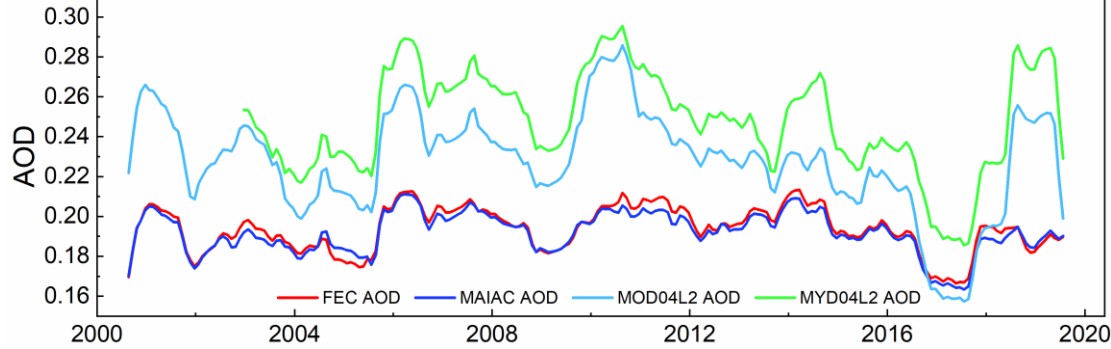


Figure 5. The long-term change trends of four AOD products obtained by removing
seasonal cycles.

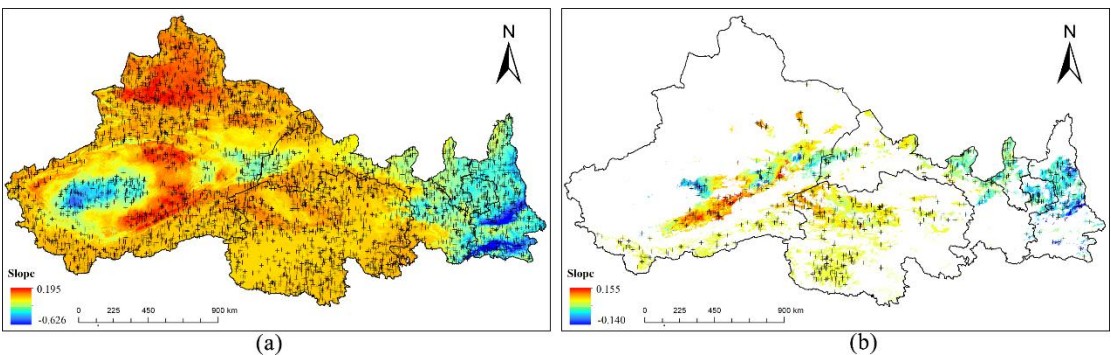


Figure 6. Spatial patterns of AOD trends obtained by removing seasonal cycles between

2000 and 2019: (a) FEC AODs ($10^{-3}$) and (b) MAIAC AODs ($10^{-2}$). The label '+'

indicates a statistically significant trend ($p < 0.05$).

As is well established, the effect of scale is a scientific problem in remote sensing,

so we further discussed the ability of the FEC AOD product to describe relatively fine-

spatial-resolution features. First, we created a 10 km×10 km fishnet; then, we chose a

single LUCC as a corresponding ecosystem; and finally, we selected five different

ecological zones (forest, grassland, farmland, construction land, and unused land) to

further quantify the local performance of the FEC AOD product (Figure 1). We found

that the FEC AOD and MAIAC AOD products revealed good consistency in their long-

term trends, while the MxD04L2 AODs showed a larger deviation (Figure 7). The FEC

AODs and MAIAC AODs had close relationships in unused lands (R = 0.959) and

farmland, followed by in construction land and forest, while these products had the

lowest relationship in grassland (R = 0.675). Therefore, the above evidence indicates

that the FEC AOD product is also reliable with regards to the fine-spatial-resolution

long-term trends captured over single surface coverage types.

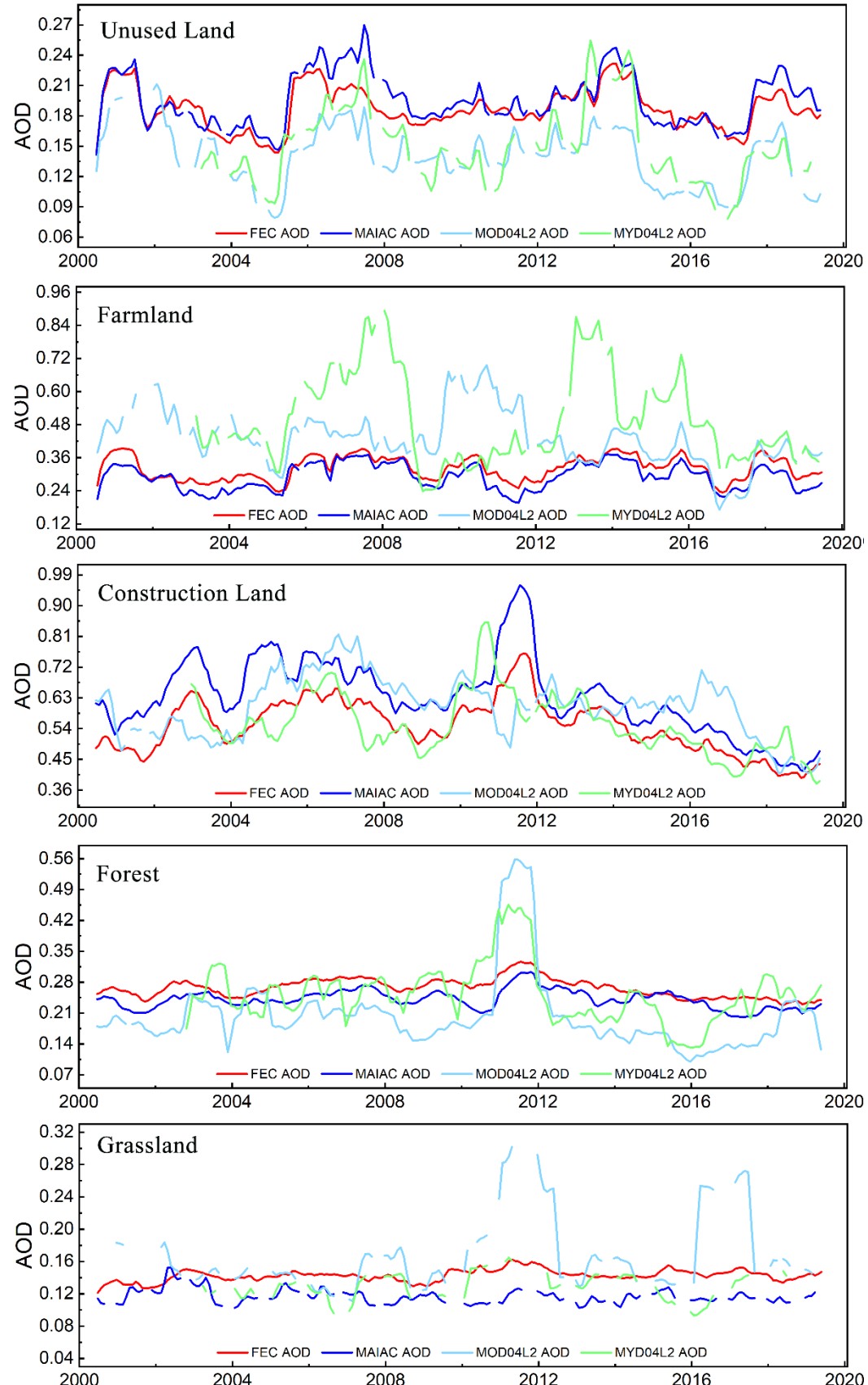

431

432    Figure 7. The long-term change trends of four AOD products over five ecological zones

433    obtained by removing seasonal cycles.

## *3.3 Spatiotemporal patterns of the FEC AOD product from 2000 to 2019*

Generally, the spatial patterns of the FEC AOD product were consistent among different years (Figure S5); the highest AODs were found in the southern area of Xinjiang Uyghur Autonomous Region of China (hereafter referred to as Xinjiang) and the centre of Shaanxi Province, mainly due to the special meteorological conditions, unique topography and surface coverage of these regions. The AODs were low in other areas, especially in southern Qinghai Province. The multiyear mean AOD was $0.193 \pm 0.124$ for the whole study area. The spatial AOD patterns differed greatly at the seasonal level (Figure S4). In autumn, the AODs were relatively small, with an average AOD value of $0.147 \pm 0.089$ and most AOD values < 0.2. In contrast, the AODs were most severe in spring, with most AOD values > 0.2 (average = $0.267 \pm 0.200$). The summer and winter seasons had similar spatial patterns, and the former had higher AODs than the latter, with AOD values of $0.198 \pm 0.134$ and $0.159 \pm 0.103$, respectively. To further investigate the spatiotemporal variations in the AOD, concepts of information entropy were introduced: temporal information entropy (TIE) and time-series information entropy (TSIE) (Ebrahimi et al., 2010). TIE and TSIE are time-series indicators that can depict the changing intensity and trend information of AODs. Generally, the higher (lower) the TIE is, the stronger (weaker) the change intensity of the AOD is in the temporal dimension. Regarding TSIE, if TSIE >0, then the AOD is increasing in the considered period, whereas TSIE <0 denotes a downward trend. Furthermore, the larger the absolute TSIE value is, the more significant the increasing (decreasing) trend is. Figure 8 depicts the TIE and TISE of the AOD from 2000 to 2019 over the whole study area. We find that the overall AOD change intensity over the past 20 years is large, especially in southern Xinjiang (Taklimakan Desert) and Shannxi Province (Loess Plateau). The areas with low variation intensities are distributed mainly at high elevations (in mountainous and grassland areas). The change intensity characteristics are similar to the AOD changes, meaning that the higher the AOD is, the larger the multiyear change is. The AODs in Xinjiang were increasing throughout the study period, with the most obvious increases occurring around the Taklimakan Desert and northern

Xinjiang, whereas the eastern AODs were decreasing, with decreases concentrated
mainly in Shannxi Province and south-eastern Gansu Province. Considering TIE and
TSIE together, we find that the AODs increased strongly in the south-eastern
Taklimakan Desert but increased slightly in northern Xinjiang and north-western
Qinghai Province. The AODs in southern Qinghai Province showed almost no change.
A dramatic decrease was observed found in the eastern area, mainly distributed in
Shannxi Province, Ningxia Hui Autonomous Region, and south-eastern Gansu
Province. A possible reason for this finding is that the Loess Plateau is experiencing
greening, and the vegetation cover in this region continues to increase under artificial
intervention. All these various characteristics are in good agreement with the detrended
long-term variation results (Figure 6).

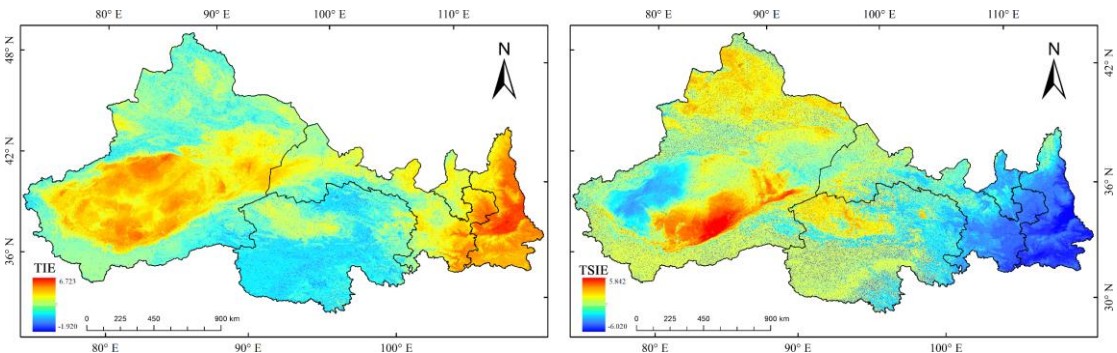

Figure 8. Temporal information entropy (TIE) and time-series information entropy
(TSIE) of the AOD distribution.
The FEC AOD product, with its high spatial resolution and full coverage over arid
and semi-arid areas, provides new possible data sources for further fine-scale research
on air pollution in areas with scarce data. Based on the FEC AOD product, we explored
the regional distribution characteristics of AODs under different areas and surface
coverage types. Figure 9 shows that the AODs in Gansu Province were highest in all
months, while the AODs in Qinghai Province were lowest. From January to December,
almost all AODs showed a trend of first increasing and then decreasing, peaking in
March and April. Except for Gansu Province, where the AODs were bimodal, the other
provinces/autonomous regions exhibit unimodal AODs. Figure 10 describes the
seasonal AOD distribution under seven different land cover types (forest, grassland,
waterbody, ice and snow, construction land, unused land, and farmland). The AODs
over ice and snow were the smallest and continuously decreased from spring to winter.
The AODs were at high levels over farmland and construction land, mainly due to
human activities. Regardless of the land cover type, the springtime AODs were always
highest. Except for ice and snow and unused land, the seasonal AOD distributions were
similar among land cover types, first decreasing and then increasing, and autumn had
the lowest AOD values.

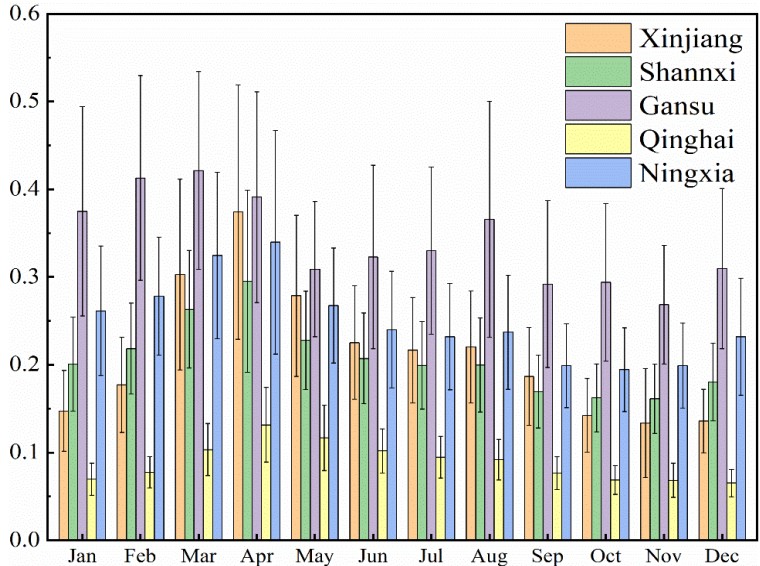


Figure 9. The monthly AOD distribution characteristics in different
provinces/autonomous regions. The error bars represent the standard errors.

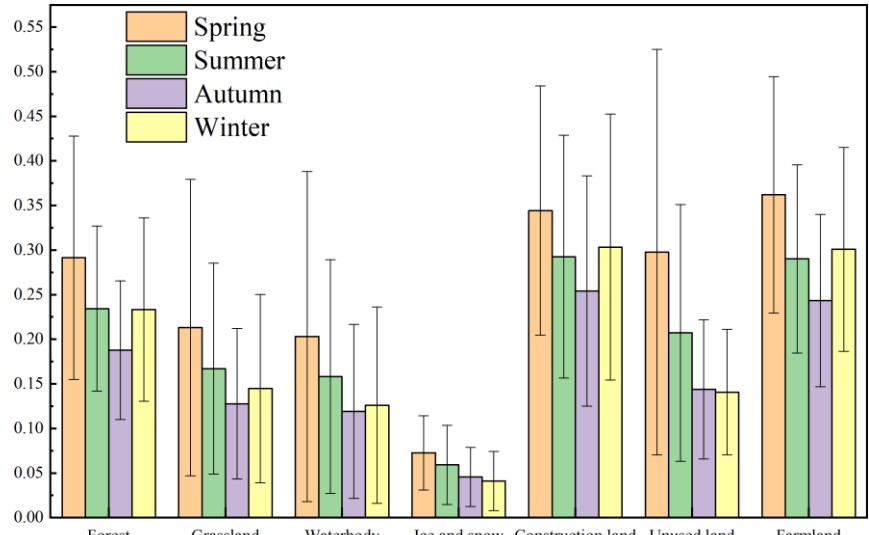


Figure 10. The seasonal AOD distributions over different land cover types. The error
bars represent the standard errors.

## 3.4 Variation partitioning of FEC AOD

To examine the contributions of environmental covariates to the FEC AOD
dynamics, redundancy analysis (RDA) was used to explore the association between
different seasons in the FEC AOD product and environmental covariates. The twelve
environmental covariates were divided into three groups: meteorological parameters,
surface properties, and terrain factors. The variance proportions driving the FEC AOD
variations on different temporal scales were tested from the environmental covariate
groups. The variation in the FEC AODs can be interpreted by every group of
environmental covariates individually or using the combined variation owing to a set
of two or more covariates, and the residual represents the unexplained proportion. The
variance partitioning results can be described as Venn diagrams constructing in the R
language (Waits et al., 2018). From Table 2 and Figure 11, the variation partitioning
analysis reveals that the meteorological factors still explain a maximal proportion of
FEC AOD variance on different temporal scales, followed by the terrain factors, and
the surface properties explain the smallest variation; the average contributions of these
categories were 77.1%, 59.1%, and 50.4%, respectively. In different seasons, the
environmental covariates have different abilities to explain the FEC AODs, and the
following order was obtained: winter (86.6%) > autumn (80.8%) > spring (79.9%) >
summer (72.5%). Except in winter, the largest variance was explained by the three
groups' environmental covariates, with values of 40.7%, 38.9%, and 45.4% in spring,
summer, and autumn, respectively. In winter, the largest variance was explained by
meteorological and terrain factors (39.1%). From spring to winter, the explanatory
ability of the three groups of covariates was always highest in autumn, and
meteorological parameters, surface properties, and terrain factors reached their lowest
values in summer, winter, and spring, respectively.

Table 2. Three groups of environmental covariates for AOD variation partitioning

| Variance proportion | Spring | Summer | Autumn | Winter | Average |
| --- | --- | --- | --- | --- | --- |
| Meteorological parameters | 78.8% | 70.4% | 80.5% | 74.8% | 77.1% |
| Surface properties | 44.5% | 37.9% | 52.5% | 31.4% | 50.4% |
| Terrain factor | 48.7% | 49.5% | 62.6% | 62.8% | 59.1% |
| Residual | 20.1% | 27.5% | 19.2% | 13.4% | 21.8% |


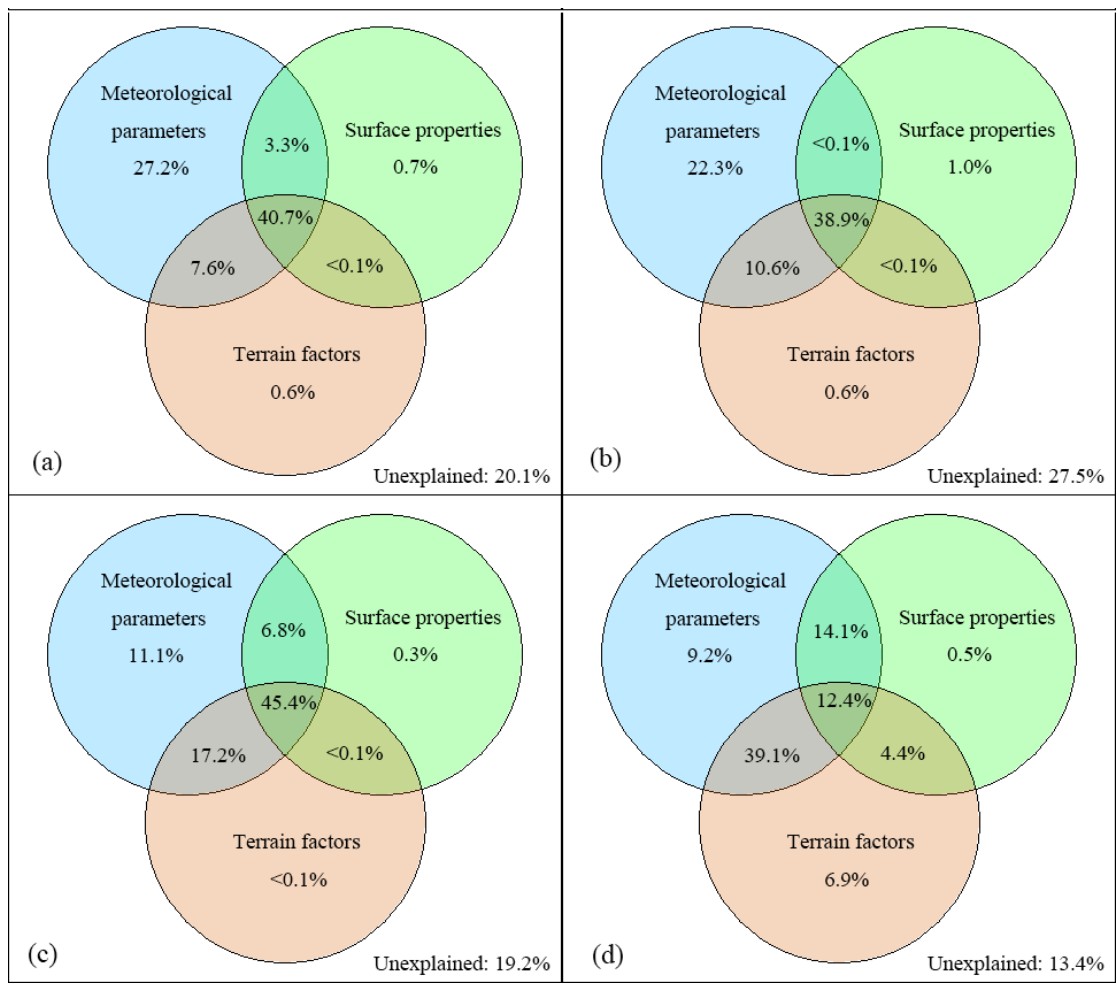

Figure 11. Seasonal variation partitioning and average AODs explained in the following seasons: (a) spring; (b) summer; (c) autumn; and (d) winter.

## 4 Discussion

### 4.1 Model uncertainty

This study, based on the MAIAC AOD product and 12 environmental covariates data, adopts the bagging tree ensemble approach to produce a monthly advanced-performance, full-coverage, and high-resolution FEC AOD product in northwest China. The bagging tree ensemble approach has a strong advantage in feature modelling and prediction, but some problems also exist; for example, most base learners are black boxes, meaning that their explanation capacities are limited (Zounemat-Kermani et al., 2021). Concurrently, the model uncertainty is also an issue to be considered, possibly

arising from the setting of hyperparameters and base learners and the sample number
selection (Khaledian and Miller, 2020). Therefore, the model robustness is critical for
modelling and predicting. Simultaneously, providing mapping uncertainty information
helps users better understand the quality of the FEC AOD product in different regions,
thus further promoting users' reasonable use of the AOD product. To check the
reliability and stability of the simulated AOD model and consider the computing
efficiency simultaneously, data representing one month were randomly selected
(August 2010), and we conducted a 100-iteration, 10-fold cross-validation; that is, we
obtained 100 predictions for each pixel and averaged these 100 predictions to obtain
the final prediction result (Rodriguez et al., 2010; Wei et al., 2021; Zhang et al., 2021;
Ma et al., 2022). Then, we calculated the model uncertainty, specifically by using the
standard deviation and upper and lower limits of the 95% confidence interval (Text S1).
Following the 100 experiments, the validated $R^2$ value still remained at 0.90, and the
RMSE and MAE values ranged from 0.058-0.057 and 0.0319-0.0317, respectively.
Concurrently, the case average and uncertainty results are shown in Figure 12. The FEC
AODs were concentrated mainly in the 0-0.6 range, accounting for 96.2% of all AODs,
and the maximum distribution was 0.1-0.2 (36.8%). The uncertainty was concentrated
mainly within the 0.2-0.6 range, accounting for 80.0%, and the maximum distribution
was 0.4-0.5 (38.1%). We also calculated the average uncertainties corresponding to
different FEC AOD levels (Figure 13). Uncertainties below 0.5 accounted for 77.3% of
the region, and the lowest uncertainty (0.3) corresponded to the largest proportion of
the FEC AODs (0.1-0.2). With increasing AODs, the uncertainty also continued to rise;
in other words, the high-AOD areas often featured high uncertainty.

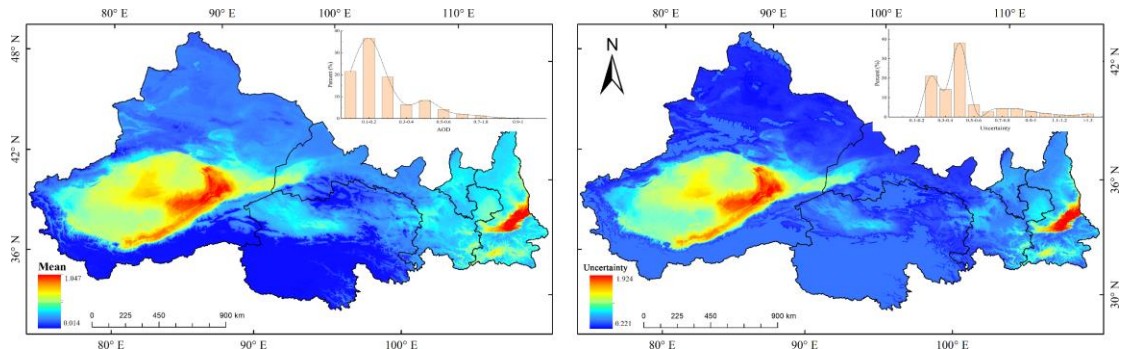


Figure 12. Distributions of the mean values and uncertainties in the AOD prediction
model.

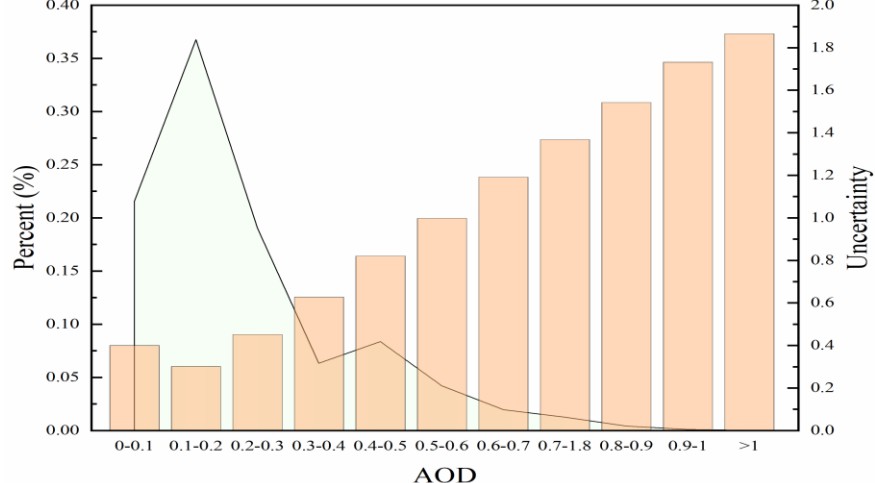

Figure 13. The average uncertainties corresponding to different AOD levels. The light-
coloured area surrounded by the black line denotes the AOD percentage, and the
histogram bars reflect the uncertainty.
*4.2 AOD as affected by environmental covariates*
The bagging tree ensemble method performance is generally affected by the
selection of environmental covariates (Khaledian and Miller, 2020). The prediction
accuracy is dependent on input variables, with underpinning static variables, and
meteorological factors (dynamic variables) explain most of the AOD variation (Yan et
al., 2022). Despite our selection of 12 environmental covariates that could explain most
of the AOD variations, approximately 13.4% - 27.5% of the results could not be well
explained, and differences in the interpretation of the environmental covariates existed.
Therefore, there is much room for improvement in the optimization of environmental
covariates. There is no doubt that the meteorological parameters are the most significant
contributors, as the temperature, precipitation, evapotranspiration, and wind speed
effectively influence the AOD through direct or indirect interactions in the air (Chen et
al., 2020). At the same time, the effect of terrain factors cannot be ignored, as these
factors affect the propagation, diffusion, and settlement of the AOD. The surface factors
involving the surface cover and soil wetness affect the dust generation and reduction
processes. Additionally, some other questions also warrant further research, such as
surface properties performance to explain AOD in summer lower spring and examining
why the terrain factors have a higher AOD variance analytical power in autumn and
winter compared to in spring and summer. We preliminarily speculate that this may be
related to multifactor interactions, but this topic needs further analysis. In the following
research, we consider introducing more related environmental covariates to try to
improve the prediction accuracy. In addition, we plan to further explore the internal
correlations among various covariates and the relative contributions of individual
covariates to the AOD. Of course, the high spatial resolution and accuracy of the
environmental covariates must also be taken into consideration (by adding or replacing
data as necessary).
*4.3 FEC AOD product for local information characterizations over*
*complex underlying surfaces*
Spatial heterogeneity, as the 2nd law of geography, is the source of the scale effects.
As a result, the richness of feature information varies in accordance with spatial scales
in remote sensing data; in most cases, certain patterns are found only at specific scales
(Miller et al., 2015). Complex underlying surfaces are often accompanied by strong
spatial heterogeneity and scale effects, which bring great challenges to high-spatial-
resolution remote sensing observations and product generation. In this research, the
FEC AOD product, which is generated by the way in which MAIAC AOD is
constrained by combining dynamic and static variables, was consistent with the MAAC
AOD product overall. Specifically, the monthly correlations were all above 0.78 in the
study area, and most were higher than 0.9 (N = 240, Rmean = 0.928, P < 0.001, Figure
S3). In addition, the FEC AOD product was also found to be reliable in fine-resolution,
long-term trend captures performed on single ecosystems. However, the performance
of the FEC AOD product on complex surfaces needs further exploration. Two typical
cities (Urumqi and Lanzhou) and two months (April and October) were randomly
selected to analyse the applicability of the FEC AOD product over complex underlying
surfaces, and Shaybak District and Chengguan District were randomly selected for
magnification in the cities of Urumqi and Lanzhou, respectively (Figure 14). Obviously,
the MOD04L2 and MYD04L2 AOD products were not suitable for local air quality
research because it is difficult to characterize the detailed features of AODs using these
products due to their coarse spatial resolutions and excessive nodata values. However,
we also identified some evident differences between the FEC AOD and MAIAC AOD
products, especially in April of 2010 over south-eastern Urumqi. To this end, we
quantitatively analysed the difference between the FEC AOD and MAIAC AOD
products in April 2010 over Urumqi (Figure S9). The FEC AOD and MAIAC AOD
products were similar in the north-western region (±0.05, close to the magnitude of one
standard deviation) but were obviously different in the south-eastern area. Accordingly,
we carefully compared multiple AOD products in April 2010 over Urumqi to attempt
to identify the reasons for this evident difference and determine its rationality. From
Figure 15, we found significant heterogeneities in some areas, and the portrayal of local
AOD features varied from product to product; for example, the FEC, MERRA-2,
MERIS, MOD04L2, and MOD08 AOD products showed high values in south-eastern
Urumqi. Therefore, we think that the main reasons for the evident difference between
the FEC AOD and MAIAC AOD products in south-eastern Urumqi may be as follows:
(1) Algorithm limitations. The MAIAC algorithm assumes that the surface state is stable
over a short period of time, resulting in a large number of high AOD records not being
detected in the MAIAC AOD product (Lyapustin et al., 2018; Lyapustin et al., 2011).
Certainly, our model and our selection of environmental covariates also introduce some
uncertainty, as was systematically discussed above. (2) Scale effects and spatial
heterogeneity. Scale effects are common phenomena in remote sensing and are
inevitable and difficult to eliminate. When scale effects overlay spatial heterogeneity,
it may be difficult for the AOD retrieval algorithm to process data under the existing
technology level. In this situation, most modes may have fuzzed or smoothed AOD
extrema and thus cannot effectively capture local information. Despite the significant
differences in April 2010 over south-eastern Urumqi, we found that the FEC AOD
product still has a good ability to capture long-term trends in Urumqi (Figure S10-S11).
The FEC AOD and MAIAC AOD products have a close relationship in Midong District
($R = 0.811$) and Dabancheng District, while these products have the lowest relationship
in Shaybak District ($R = 0.620$). In summary, the evident differences between the FEC
AOD and MAIAC AOD products in some highly heterogeneous areas are objective and
reasonable in some way, but there is still much research to be done to determine which
AOD products are the most reliable in portraying local features.

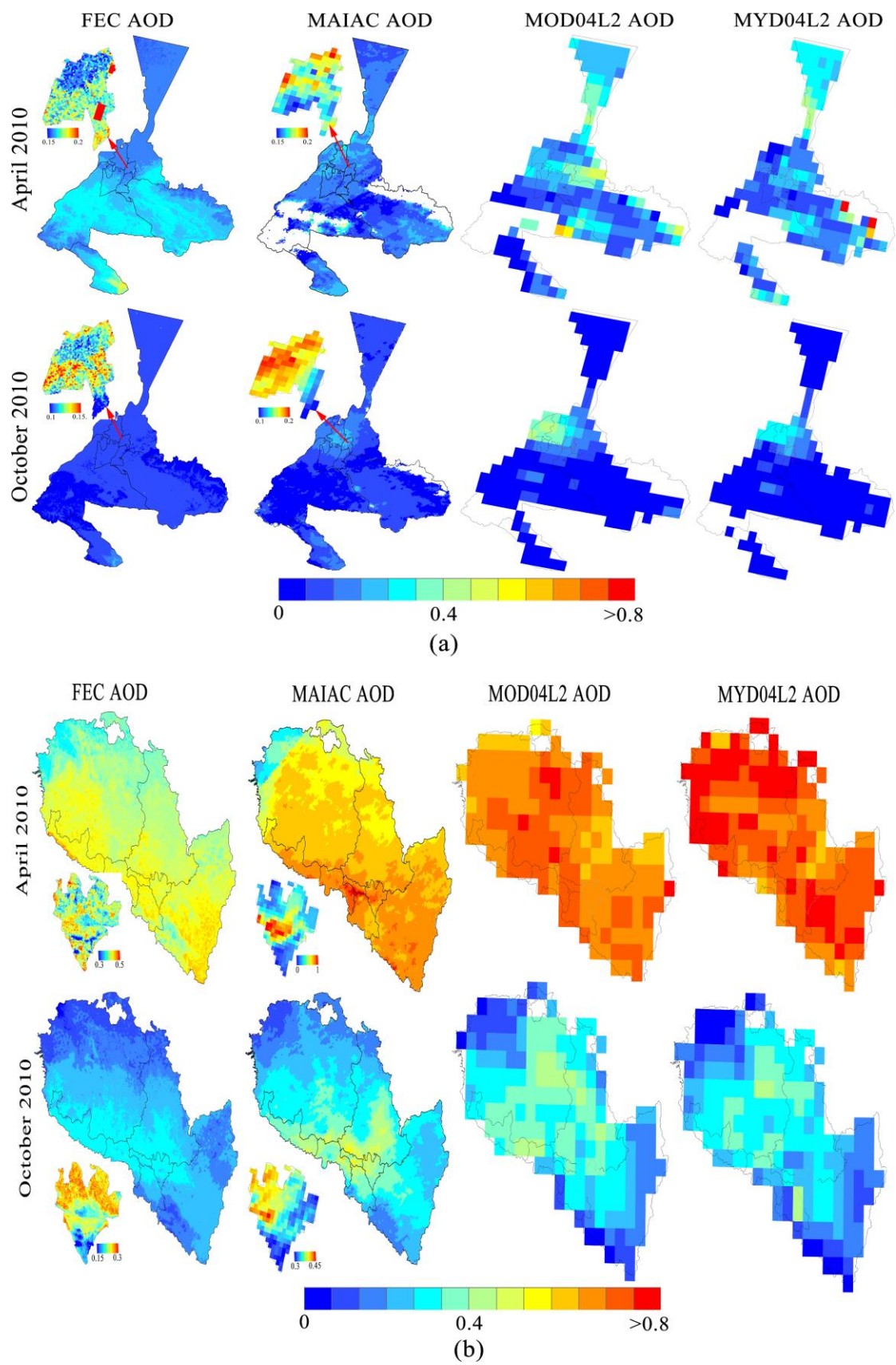

Figure 14. The spatial patterns of four AOD products in April and October 2010: (a)
Urumqi and (b) Lanzhou.

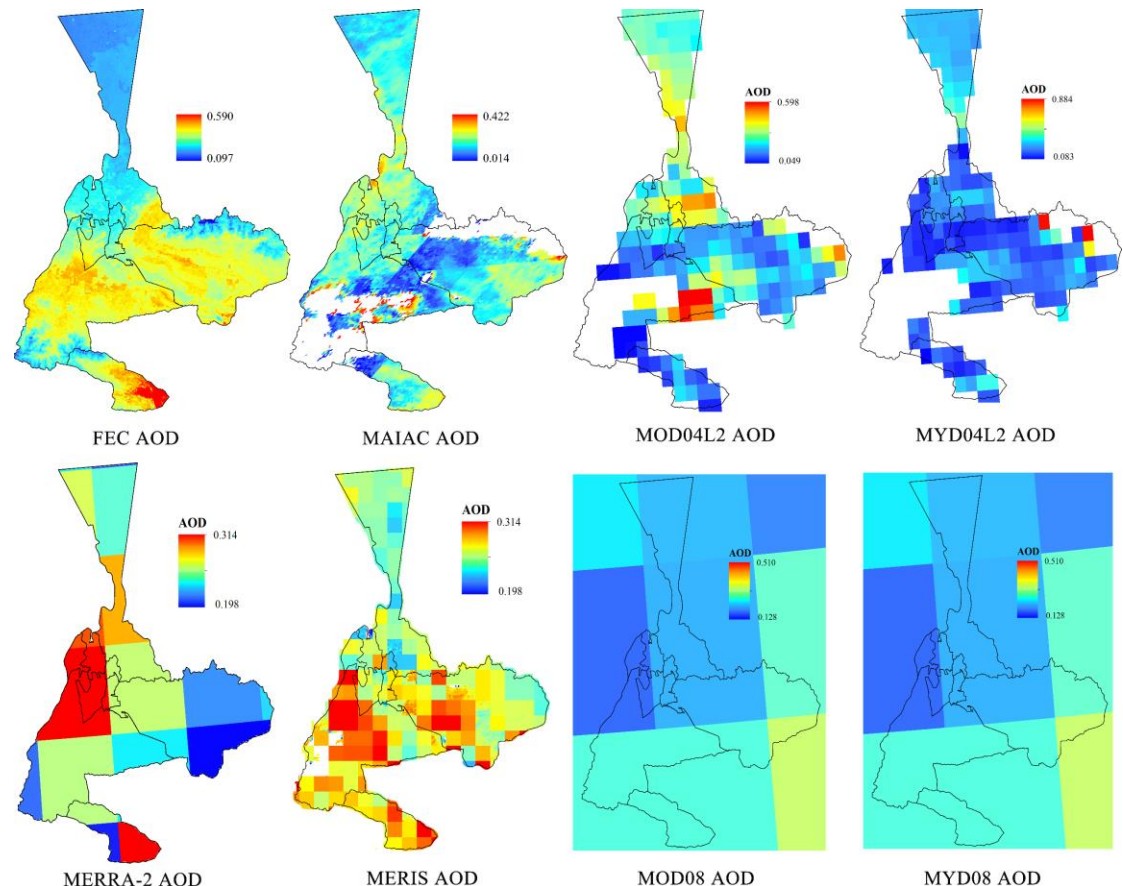


Figure 15. The spatial patterns between the FEC AOD product and other AOD products
in April 2010 over Urumqi.

## 5 Data availability

This monthly advanced-performance, full-coverage, high-resolution AOD dataset
(FEC AOD) constructed over northwest China in this study is freely available via
https://doi.org/10.5281/zenodo.5727119 (Chen et al., 2021a).

## 6 Conclusion

In this paper, a monthly advanced-performance, full-coverage, high-resolution
AOD dataset was produced based on the MAIAC AOD product and multiple
environmental covariates and utilizing a machine learning method from 2000 to 2019
in the northwest region of China. AERONET and MODIS AOD data were collected to
verify the applicability of the FEC AOD product. Then, the spatiotemporal changes
reflected in FEC AOD product are analysed, and an interpretation of the contributions
of environmental covariates to the FEC AOD product is explored. The results show that
the FEC AOD effectively compensates for the deficiency and constraints of in-situ
observation and satellite AOD products. Moreover, the FEC AOD product demonstrates
a reliable performance and ability to capture local information and long-term trends,
even superior to the abilities of the MAIAC and MxD04L2 AOD products; these
findings also indicate the necessity of high-spatial-resolution AOD data. The spatial
patterns are consistent among different years and greatly differ at the seasonal level.
The higher the AOD is, the stronger the temporal variability is. The AODs exhibit a
dramatic decrease on the Loess Plateau and an evident increase in the south-eastern
Taklimakan Desert between 2000 and 2019. Farmland and construction land have high
AOD levels compared to other land cover types. The meteorological factors
demonstrated the maximum interpretation ability of AODs on all analysed temporal
scales, while the capability of environmental covariates to explain AODs varies
seasonally.

**Author contribution:** Xiangyue Chen designed and developed the methodology and software,
conducted the analysis and validation, and wrote the paper. Hongchao Zuo supported and supervised
the study. Zipeng Zhang developed the methodology and reviewed the paper. Xiaoyi Cao and Jikai
Duan investigated and developed the methodology. Chuanmei Zhu and Zhe Zhang performed the
conceptualization of and investigations in this study. Jingzhe Wang supported and supervised the
study and reviewed the paper.

**Competing interests.** The authors declare that they have no conflicts of interest.

**Acknowledgements:** This work was jointly supported by the Second Tibetan Plateau Scientific
Expedition and Research Program (STEP) (Grant No. 2019QZKK0103), Basic Research Program
of Shenzhen (No. 20220811173316001), Guangdong Basic and Applied Basic Research Foundation
(No. 2020A1515111142) and Key Laboratory of Spatial Data Mining & Information Sharing of
Ministry of Education, Fuzhou University (No. 2022LSDMIS05). We are grateful to the
Atmosphere Archive and Distribution System (https://search.earthdata.nasa.gov) and AERONET
(http://aeronet.gsfc.nasa.gov) for providing extensive data support for our research.

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
