# Peer review of "Full-coverage 250 m monthly aerosol optical depth dataset"

_Earth System Science Data, 2021_

## Author Comment (AC1)

July 19, 2022

**The Editor Earth System Science Data**

Dear David Carlson

On behalf of my co-authors, we thank you very much for giving us the opportunity to revise the manuscript titled "Full-coverage 250 m monthly aerosol optical depth dataset (2000-2019) emended with environmental covariates by the ensemble machine learning model over the arid and semi-arid areas, NW China" (Manuscript ID: essd-2021-426). Also, we are grateful to two anonymous reviewers for their recognition and constructive comments and feedback on our manuscript.

Based on the reviewers for their attentive and insightful reviews of the manuscript, we have made the appropriate corrections and clarifications to the manuscript, especially with regard to FEC AOD generation and validation. We have performed a more rigorous and methodological analysis and repeated the experiments with comprehensive validation. Revised portions are marked in red and highlighted in the manuscript. The responses to each of the points raised by the reviewers are also in red after each of their comments.

We hope that with these revisions the manuscript warrants full acceptance for publication.

Thank you for the opportunity and for further considering the publication of this manuscript.

Sincerely,

Hongchao Zuo Xiangyue Chen

College of Atmospheric Sciences Lanzhou University Lanzhou China Email: zuohch@lzu.edu.cn; chenxy20@lzu.edu.cn

**Responses to RC1**

Review of Full-coverage 250 m monthly aerosol optical depth dataset (2000-2019) emended with environmental covariates by the ensemble machine learning model over the arid and semi-arid areas, NW China By Chen et al. This manuscript applies bagging trees ensemble methods to produce monthly full-coverage and high-resolution AOD product (FEC AOD). Compared with AERONET AOD, FEC AOD has good performance with an  $R^2$  of 0.79. A good analysis of spatio-temporal variability is then presented and the interpretation of environmental covariates on FEC AOD is explored using redundancy analysis. I would like to recommend minor revisions.

Response: Many thanks for reviewing our manuscript and providing us with your recognition and valuable advice on our work, we studied your comments and responded to them point by point carefully as described below.

 Line 33, the expression is ambiguous since the bimodal pattern usually refers to the aerosol size distribution.

Response: Thank you for your valuable suggestion. Our original intention is to express that AOD annual variation in Gansu province shows different characteristics from other provinces, where AOD has two peaks, while in other provinces it has only one. Regarding the expression of bimodal and unimodal, we also referred to the previous studies before expressing it in this way. Of course, to avoid ambiguity, we phrased it in the revision as "the AOD annual variation pattern shows a different feature, with two peaks in March and August respectively over Gansu province, but only one peak in April over other provinces." (Page 2/Lines:30-32).

2. The blank space before the reference is lacking.

Response: Thank you for your careful reading. In the revision, we carefully checked the full text to make sure there were spaces before each reference.

**3.** How did you get the FEC AOD at 250m resolution? Is it simply a matter of interpolating the original input data to a resolution of 250m and then inputting it into the model to get the FEC AOD?

Response: Thank you for your precious question and comment. FEC AOD is built by a downscaling method, not by simple interpolation. Actually, the basic idea for downscaling AOD with bagging trees ensemble machine learning (ML) models is to train the relationships between MAIAC AOD and the auxiliary environmental variables at coarse resolution (1 km) using ML algorithms. We then apply the trained relationships to generate a high-resolution FEC AOD product at a fine resolution (250 m). This idea of downscaling has been developed more maturely and is widely used[1-3], and it is based on a complex mathematical feature that is capable of mining the characteristics of different environmental auxiliary variables on the representation of AOD. Compared with the traditional model with poor data mining ability, low accuracy, and coarse spatial resolution, the ML approach is noise-resistant and can effectively reduce modeling variance to improve model accuracy and build robust relationships between AOD and environmental auxiliary variables. In terms of auxiliary

environmental variables, we adopt a high resolution (< 250 m, i.e. 30 m or 90 m) to describe static variables, while for dynamic variables, a spatial scale of 1 km is used whenever possible. As for static variables, we only use resample to 250 m and 1 km (for LUCC, use the nearest neighbor method, and others employ the bilinear method). In terms of dynamic variables, firstly, for the ET and NDVI data below 1 km resolution, we downscaled them to 1 km using the Cubist downscaling method, not by a simple interpolation[3]. What is more, the environmental variables we have chosen are also closely related to AOD, affecting AOD production, diffusion, reaction, and sedimentation, so that the prediction of AOD can achieve better results.

- Duveiller, G., Filipponi, F., Walther, S., Köhler, P., Frankenberg, C., Guanter, L., and Cescatti, A.: A spatially downscaled sun-induced fluorescence global product for enhanced monitoring of vegetation productivity, Earth Syst. Sci. Data, 12, 1101–1116, https://doi.org/10.5194/essd-12-1101-2020, 2020.
- [2] Yang, Q., Yuan, Q., Li, T., and Yue, L.: Mapping PM2.5 concentration at high resolution using a cascade random forest based downscaling model: Evaluation and application, Journal of Cleaner Production, 277, 123887, https://doi.org/10.1016/j.jclepro.2020.123887, 2020.
- [3] Ma, Z., Shi, Z., Zhou, Y., Xu, J., Yu, W., and Yang, Y.: A spatial data mining algorithm for downscaling TMPA 3B43 V7 data over the Qinghai–Tibet Plateau with the effects of systematic anomalies removed, Remote Sensing of Environment, 200, 378-395, https://doi.org/10.1016/j.rse.2017.08.023, 2017.
- In Figure 3, please include a monthly comparison of MAIAC AOD with AERONET AOD for the same period.

Response: Thank you for your valuable advice. In the revision, we added the monthly comparison of MAIAC AOD with AERONET AOD for the same period. In addition, considering RC2 comments, we also added the monthly comparison of the MODIS 10 km AOD product (MOD04L2 and MYD04L2) with AERONET AOD for the same

---

## Author Comment (AC2)

**Responses to RC1**

Review of Full-coverage 250 m monthly aerosol optical depth dataset (2000-2019) emended with environmental covariates by the ensemble machine learning model over the arid and semi-arid areas, NW China By Chen et al. This manuscript applies bagging trees ensemble methods to produce monthly full-coverage and high-resolution AOD product (FEC AOD). Compared with AERONET AOD, FEC AOD has good performance with an $R^2$ of 0.79. A good analysis of spatio-temporal variability is then presented and the interpretation of environmental covariates on FEC AOD is explored using redundancy analysis. I would like to recommend minor revisions.

Response: Many thanks for reviewing our manuscript and providing us with your recognition and valuable advice on our work, we studied your comments and responded to them point by point carefully as described below.

1. Line 33, the expression is ambiguous since the bimodal pattern usually refers to the aerosol size distribution.

Response: Thank you for your valuable suggestion. Our original intention is to express that AOD annual variation in Gansu province shows different characteristics from other provinces, where AOD has two peaks, while in other provinces it has only one. Regarding the expression of bimodal and unimodal, we also referred to the previous studies before expressing it in this way. Of course, to avoid ambiguity, we phrased it in the revision as "the AOD annual variation pattern shows a different feature, with two

peaks in March and August respectively over Gansu province, but only one peak in April over other provinces." (Page 2/Lines:30-32).

**2.** The blank space before the reference is lacking.

Response: Thank you for your careful reading. In the revision, we carefully checked the full text to make sure there were spaces before each reference.

**3.** How did you get the FEC AOD at 250m resolution? Is it simply a matter of interpolating the original input data to a resolution of 250m and then inputting it into the model to get the FEC AOD?

Response: Thank you for your precious question and comment. FEC AOD is built by a downscaling method, not by simple interpolation. Actually, the basic idea for downscaling AOD with bagging trees ensemble machine learning (ML) models is to train the relationships between MAIAC AOD and the auxiliary environmental variables at coarse resolution (1 km) using ML algorithms. We then apply the trained relationships to generate a high-resolution FEC AOD product at a fine resolution (250 m). This idea of downscaling has been developed more maturely and is widely used[1-3], and it is based on a complex mathematical feature that is capable of mining the characteristics of different environmental auxiliary variables on the representation of AOD. Compared with the traditional model with poor data mining ability, low accuracy, and coarse spatial resolution, the ML approach is noise-resistant and can effectively reduce modeling variance to improve model accuracy and build robust relationships between AOD and environmental auxiliary variables. In terms of auxiliary

environmental variables, we adopt a high resolution (< 250 m, i.e. 30 m or 90 m) to describe static variables, while for dynamic variables, a spatial scale of 1 km is used whenever possible. As for static variables, we only use resample to 250 m and 1 km (for LUCC, use the nearest neighbor method, and others employ the bilinear method). In terms of dynamic variables, firstly, for the ET and NDVI data below 1 km resolution, we downscaled them to 1 km using the Cubist downscaling method, not by a simple interpolation[3]. What is more, the environmental variables we have chosen are also closely related to AOD, affecting AOD production, diffusion, reaction, and sedimentation, so that the prediction of AOD can achieve better results.

[1] Duveiller, G., Filipponi, F., Walther, S., Köhler, P., Frankenberg, C., Guanter, L., and Cescatti, A.: A spatially downscaled sun-induced fluorescence global product for enhanced monitoring of vegetation productivity, Earth Syst. Sci. Data, 12, 1101–1116, https://doi.org/10.5194/essd-12-1101-2020, 2020.

[2] Yang, Q., Yuan, Q., Li, T., and Yue, L.: Mapping $PM_{2.5}$ concentration at high resolution using a cascade random forest based downscaling model: Evaluation and application, Journal of Cleaner Production, 277, 123887, https://doi.org/10.1016/j.jclepro.2020.123887, 2020.

[3] Ma, Z., Shi, Z., Zhou, Y., Xu, J., Yu, W., and Yang, Y.: A spatial data mining algorithm for downscaling TMPA 3B43 V7 data over the Qinghai–Tibet Plateau with the effects of systematic anomalies removed, Remote Sensing of Environment, 200, 378-395, https://doi.org/10.1016/j.rse.2017.08.023, 2017.

**4.** In Figure 3, please include a monthly comparison of MAIAC AOD with AERONET AOD for the same period.

Response: Thank you for your valuable advice. In the revision, we added the monthly comparison of MAIAC AOD with AERONET AOD for the same period. In addition, considering RC2 comments, we also added the monthly comparison of the MODIS 10 km AOD product (MOD04L2 and MYD04L2) with AERONET AOD for the same

period. At the same time, we have also modified and added the relevant statements

[Figure]

Figure 3. Comparison with AERONET AOD. (a) FEC AOD, (b) MAIAC AOD, (c) MOD04L2 AOD, (d) MYD04L2 AOD. The red line denotes the regression line, the black line shows the 1:1 line, and the blue area indicates the 95% prediction interval.

**5.** In Line 351-352, the author concludes that FEC AOD products demonstrate a reliable accuracy and ability to capture local information, even superior to MAIAC and MxD08 AOD products. However, the loess-based seasonal trend decomposition procedure (STL) in Figure 5 does not show the advantage of FEC AOD over MAIAC AOD. If the advantage is only the spatial resolution, as described in the third point, wouldn't we be able to get any resolution with interpolation?

Response: Thank you for your precious advice. Firstly, in Section 3.1, we intend to verify the performance of FEC AOD based on in-situ and satellite respectively. In terms of the loess-based seasonal trend decomposition procedure (STL) in Figure 5, our starting point is to use SLT to compare the temporal consistency of the FEC AOD with other AOD products to demonstrate FEC AOD's ability to characterize aerosol temporal variations. Actually, the FEC AOD has a good consistency with other AOD products. What is more, based on RC2 comments, we make MOD08 and MYD08 transfer to MOD04 and MYD04 and find the accuracy advantage still remains. This again also supports the reliability of FEC AOD.

About "FEC AOD products demonstrate a reliable accuracy and ability to capture local information, even superior to MAIAC and MxD08 AOD products", which is a conclusion about section 3.1, that is, it is a general summary. Of course, we have also made corresponding modifications in the revision to avoid misunderstandings (Page 18/Lines:371-373).

In this paper, our advantage mainly lies in the improvement on spatial resolution with an effective downscaling method, but also filling the gap in no data areas. As we all

know, the scale effect is a classical issue in remote sensing, and many fine features still need to be revealed by high-resolution data[1,2]. So a high-resolution and accurate dataset is crucial to future research, especially in the data scarcity zone. If only from the perspective of interpolation, we can get any spatial resolution AOD in theory, but its accuracy and spatiotemporal consistency are difficult to guarantee, and the most important point is that interpolation ignores multi-environmental variables inter-relationship and intrinsic association constraints, but ML makes up for these deficiencies well. In addition, in terms of AOD, not the higher spatial resolution is better, some studies have shown that it is appropriate to study at a scale of 250-500 m[3,4]. Actually, with higher resolution of relevant environmental variables, by the effective downscaling model, we can theoretically obtain higher performance AOD, which not only advances the discipline but also fills the data gaps and narrows the knowledge gap. This study does a good trial following the above guidelines.

[1]    Atkinson, Peter M., A. Stein, and C. Jeganathan.: Spatial sampling, data models, spatial scale and ontologies: Interpreting spatial statistics and machine learning applied to satellite optical remote sensing, Spatial Statistics, 50, 100646, https://doi.org/10.1016/j.spasta.2022.100646, 2022.

[2]    Yu, Ying, Yan Pan, Xiguang Yang, and Wenyi Fan.: Spatial Scale Effect and Correction of Forest Aboveground Biomass Estimation Using Remote Sensing, Remote sensing, 14, 2828, https://doi.org/10.3390/rs14122828, 2022.

[3]    Wang, Z., Deng, R., Ma, P., Zhang, Y., Liang, Y., Chen, H., Zhao, S., and Chen, L.: 250-m Aerosol Retrieval from FY-3 Satellite in Guangzhou, Remote Sensing, 13, 920, https://doi.org/10.3390/rs13050920, 2021.

[4]    Chen, X., Ding, J., Wang, J., Ge, X., Raxidin, M., Liang, J., Chen, X., Zhang, Z., Cao, X., and Ding, Y.: Retrieval of Fine-Resolution Aerosol Optical Depth (AOD) in Semiarid Urban Areas Using Landsat Data: A Case Study in Urumqi, NW China, Remote Sensing, 12, 467, https://doi.org/10.3390/rs12030467, 2020.

**6.** Please describe in detail the calculation of AOD uncertainty (lines 498-503).

Response: Thank you for your valuable comment and careful reading. In terms of Section 4.1 Model uncertainty, we randomly select a month to check the model reliability and stability. Specifically, firstly, we do 100 repetitions of the experiment. Then, we calculate model uncertainty by the standard deviation, upper and lower limits 95% confidence interval to realize (The specific calculation formula we have added in the Support Information Text S1).

Text S1. Calculation of model uncertainty

To ensure the reliability and reasonability of the FEC AOD, we performed 100 modelings and predictions for August 2010, that is, 100 times of prediction for each pixel, and the final prediction result is the average of 100 times.

$$AOD_{mean}\ (j) = \frac{1}{n}\sum_{i=1}^{n} AOD_i\ (j)$$

Where n is the number of modeling and predictions (n = 100), $AOD_i(j)$ is the AOD predicted value of the jth pixel and ith modeling, $AOD_{mean}\ (j)$ is the predicted AOD mean of the jth pixel.

The model uncertainty is calculated as follows:

$$CI_{upper}\ (j) = \mu + 1.96 \times \frac{\sigma}{\sqrt{n}}$$

$$CI_{lower}\ (j) = \mu - 1.96 \times \frac{\sigma}{\sqrt{n}}$$

$$AOD_{uncertainty}\ (j) = \frac{\left[CI_{upper}\ (j) - CI_{lower}\ (j)\right]}{AOD_{mean}\ (j)}$$

Where $CI_{upper}\ (j)\ and\ CI_{lower}\ (j)$ are the upper and lower limits of the 95% confidence interval (CI) of the $j$th pixel respectively, $\mu$ is the jth pixel AOD mean at 100 predictions, $\sigma$ is the jth pixel AOD standard deviation predictions, and n is the number of samples, $AOD_{uncertainty}\ (j)$ is the uncertainty of the $j$th pixel prediction.

---

## Author Comment (AC3)

**Responses to RC2**

High-resolution with full spatial coverage AOD dataset is important for air pollution-related studies, especially for bright surfaces. The authors have done a lot of work to generate a 250 m AOD dataset in arid and semi-arid areas in northwestern China. However, my biggest concern is the spatial resolution of current developed dataset since it is not clear how to generate the 250 m AOD dataset with two much coarse-resolution MAIAC AOD (1 km) and MxD08 (1 degrees) as main predictions although there are some high-resolution auxiliary data. It sounds incredible and I don't find any downscaling approach or descriptions in the paper.

Response: Many thanks for reviewing our manuscript and providing us with constructive feedback. We are sorry for not having expressed the downscaling method clearly in the manuscript, and this has been added in the revision of the relevant (Page 13/Lines:278-286). Actually, the 250 m AOD is generated based on the MAIAC AOD and auxiliary environmental data, and the MxD08 AOD is only used to validate the FEC AOD accuracy and temporal consistency. About downscaling approach (Section 2.6), we adopt the ensemble machine learning methods (bagging trees ensemble).

The idea behind the downscaling method is to establish either a statistical correlation or a physically-based model between coarse-scale image/product and fine-scale auxiliary variables. But a common premise for downscaling methods is that high-resolution auxiliary variables are indispensable. For AOD, elevation (DEM) and land cover (LUCC) data, which are usually at sub-kilometers resolution and have a great impact on AOD distribution, can be regarded as potential high-resolution auxiliary

variables. This ensures the feasibility of downscaling AOD products. Besides, high-resolution LUCC and DEM data are easily accessible worldwide, which contributes to the large-scale implementation of the high-resolution AOD mapping based on the downscaling method[1]. The basic idea for downscaling AOD with bagging trees ensemble machine learning (ML) models is to train the relationships between MAIAC AOD and the auxiliary environmental variables at coarse resolution (1 km) using ML algorithms. We then apply the trained relationships to generate a high-resolution FEC AOD product at a fine resolution (250 m). This idea of downscaling has been developed more maturely and is widely used[1-3], and it is based on a complex mathematical feature that is capable of mining the characteristics of different environmental auxiliary variables on the representation of AOD. At the same time, we have studied your comments and responded to them point by point carefully as described below.

[1] Duveiller, G., Filipponi, F., Walther, S., Köhler, P., Frankenberg, C., Guanter, L., and Cescatti, A.: A spatially downscaled sun-induced fluorescence global product for enhanced monitoring of vegetation productivity, Earth Syst. Sci. Data, 12, 1101–1116, https://doi.org/10.5194/essd-12-1101-2020, 2020.

[2] Yang, Q., Yuan, Q., Li, T., and Yue, L.: Mapping $PM_{2.5}$ concentration at high resolution using a cascade random forest based downscaling model: Evaluation and application, Journal of Cleaner Production, 277, 123887, https://doi.org/10.1016/j.jclepro.2020.123887, 2020.

[3] Ma, Z., Shi, Z., Zhou, Y., Xu, J., Yu, W., and Yang, Y.: A spatial data mining algorithm for downscaling TMPA 3B43 V7 data over the Qinghai–Tibet Plateau with the effects of systematic anomalies removed, Remote Sensing of Environment, 200, 378-395, https://doi.org/10.1016/j.rse.2017.08.023, 2017.

Below are some other specific comments:

**1.** The authors are suggested to summarize previous published studies focusing on multi source AOD dataset fusion or AOD gap filling using different models to rich the Introduction since a lot of related work have been done.

Response: Thank you for your careful reading and valuable suggestions. Actually, we have summarized the common methods for multi-source AOD dataset fusion or AOD gap filling in the introduction (Pages 3-4/Lines:79-94). Based on your suggestion, we have further highlighted downscaling methodology, including fusion and gap filling. In addition, we have also added some specific case references and further described the implementation of the downscaling method used in this study (Pages 3-4/Lines:85-94).

2. Section 2.3: MxD08 AOD product is too coarse in the spatial resolution (1 degrees) to be used for comparison in such a small study region. I suggest using the MxD04 product with a high resolution 3 or 10 km.

Response: Thank you for your careful reading and precise advice. According to your suggestion, we select the MxD04L2 10 km data to replace the MxD08 AOD in revision, because we have found experimentally that MOD04 _3k is generated based on the dark target method, and not applicable in this study area (most of the area is almost no data). Responding modifications are given in the text (Page 7/Lines:164-178).

*2.3 MODIS MxD04L2 data*

MYD04L2 and MOD04L2 are the level 2 atmospheric aerosol products from Aqua and Terra respectively, where spatial and temporal resolutions are 10 km × 10 km and

1 day respectively (Chen et al, 2021b). The MxD04L2 AOD product mainly provides two algorithms, the Dark Target (DT) and Deep Blue (DB) algorithms, to retrieve global AOD distribution. Based on the MODIS Collection 6.1, we chose 550 nm combined DT and DB AOD to validate FEC AOD. It is worth noting that the Aqua and Terra launch time is different, so we can acquire MOD04L2 data from March 2000 to February 2020, but as for MYD04L2, we only acquire data from July 2002 to February 2020. All processes are realized through downloading from NOAA website (https://ladsweb.modaps.eosdis.nasa.gov/) and calculating and analyzing local computer, and main works, including geometric correction, projection conversion, image mosaics, clipping, computing daily and monthly mean of AOD, and numerical extraction, perform in MODIS Reprojection Tool (MRT) and ENVI and ArcGis software.

3. Section 2.4: Please clarify the version and level of AERONET data, and the number of the stations (also suggest adding them in Figure 1) used in the study. In addition, the author should highlight the novelty of their study and the differences compared to previous studies

Response: Thank you for your careful reading and precious advice. In revision, we have clarified the version and level of AERONET data and added them in Figure 1 (Pages 6-8/Lines:144, 186-189). In terms of the novelty and differences of this study, this is the first highest resolution AOD dataset with complete coverage of Northwest China (a typical area of scarce information or limited data applicability). Secondly, the data is generated by the current mainstream machine learning algorithms, and the performance and efficiency are reliable. Of course, based on this dataset, we have also found some new phenomena, which again validate the impact of ecological

engineering on air quality in China, while raising new research questions for the southeast Taklamakan Desert. In summary, the FEC AOD effectively compensates for the deficiency and constraints of in-situ observation and satellite AOD products. Meanwhile, FEC AOD products provide a new choice for future atmosphere research in Northwest China and the ability to capture finer local information.

[Figure]

Figure 1. Study area. The figure shows typical arid and semi-arid areas, five provinces in northwest China.

**4.** Lines 273-288: It is not clearly how to train and validate the model. Please clarify what are the inputs to the model, and what is the real/true vaule for target? What are the training samples and verification samples? Are they independent of each other?

Response: Thank you for your careful reading. In this study, we use 12 environmental covariates (1 km) as downscaling method (bagging trees ensemble algorithms) input to acquire AOD-environmental covariates (AODe) model in 1 km and utilize AODe model and 250 m environmental covariates to acquire FEC AOD. In the modeling

process, we adopt the 10 cross-validation folds as a tool for verification. That is, instead of dividing the modeling set and validation set, we repeat the cross-validation. The 10 cross-validation folds take turns using 9 of them as training data to train the model and 1 as validation data to assess model performance and calculate the average validation error over all folds. This method gives a good estimate of the predictive accuracy of the final model trained using the full data set and can effectively avoid overfitting. As a result, we have no real/true value and use the pseudo instead. This again illustrates the necessity and urgency of producing high-resolution AOD datasets in this study area, which is a data-scarce area with only a few observational data. In addition, conventional methods of dividing modeling sets and validation based on truth values usually only capture the accuracy of predictions, but the 10 cross-validation folds through repeated cross-validation not only can get the accuracy but also can get the uncertainty of the prediction, for instance, knowing where the data uncertainty is higher, and then this is where we need to strengthen the observation or build stations in the future.

**5.** Figure 4: I don't see much differences compared with 1km AOD, and I think the authors need show the advantages of 250m data set, e.g., may zoom in the image by looking at the AOD distributions at urban areas.

Response: Thank you for your careful reading. Section 3.1 is to validate FEC AOD accuracy based on in-situ and satellite. Figure 4 belongs to the part of comparison with satellite AOD products. So we can find out the spatial consistency from Figure 4, and more differences between FEC AOD and MAIAC 1 km AOD, which we have also considered in the original manuscript (Pages 16-17/Lines:342-345), ie. Figure S2 and

S3. We select two typical cities (Urumqi and Lanzhou) in NW China, and randomly zoom in the districts (Shaybak and Chengguan districts) respectively, and we can find the FEC AOD has a strong potential to describe local AOD features or fine AOD distribution compared with MAIAC AOD.

[Figure]

Figure S2. The spatial pattern difference of four AOD products in April 2010 and October 2011 over Urumqi.

[Figure]

Figure S3 The spatial pattern difference of four AOD products in April 2010 and October 2011 over Lanzhou.

**6.** Lines 341-353: The results are pretty similar among different AOD products and difficult to distinguish the difference, and more quantitative comparison results are needed.

Response: Thank you for your careful reading. Actually, Section 3.1 theme is to validate FEC AOD accuracy based on in-situ and satellite. So we focus more on the consistency between the FEC AOD and other satellite AOD products. Surely, we also added some quantitative comparison results in the revision (Pages 17-18/Lines:361-370).

**7.** Sections 3.2 and 3.3: It is recommended to calculate the monthly and seasonal long-term trends and statistical significance by removing the seasonal cycles to have a look at how AOD changes throughout the study area since AOD data for nearly 20 years are available.

Response: Thank you for your valuable advice. The main objective of this study is to

create a new advanced-performance, full-coverage, and high-resolution AOD dataset and validate it. In addition, we analyze the spatiotemporal pattern in Section 3.2, and the temporal variability of AOD is deeply revealed by temporal information entropy (TIE) and time-series information entropy (TSIE), so we think this has a good look at how AOD changes throughout the study area since AOD data for nearly 20 years. Of course, we also recognize your comments, which will be further explored in the next step of FEC AOD application research.

---

## Author Response (AR2)

**The Editor Earth System Science Data**

Dear David Carlson

On behalf of my co-authors, we thank you very much for giving us the opportunity to revise the manuscript titled "Full-coverage 250 m monthly aerosol optical depth dataset (2000-2019) emended with environmental covariates by the ensemble machine learning model over the arid and semi-arid areas, NW China" (Manuscript ID: essd-2021-426). Also, we are grateful to anonymous reviewers for their careful and constructive comments and feedback on our manuscript.

Based on the reviewers for their attentive and insightful reviews of the manuscript, we have made appropriate corrections and clarifications on the manuscript, especially with regard to the comparison among FEC AOD and other AOD products in the long-term trends. We have performed a more rigorous and methodological analysis and repeated the experiments with comprehensive validation. Revised portions are marked in red and highlighted in the manuscript. The responses to each of the points raised by the reviewers are also in red after each of their comments.

We hope that with these revisions the manuscript warrants full acceptance for publication.

Thank you for the opportunity and for further considering the publication of this manuscript.

Sincerely yours,

Hongchao Zuo Xiangyue Chen

College of Atmospheric Sciences Lanzhou University Lanzhou China Email: zuohch@lzu.edu.cn; chenxy20@lzu.edu.cn

**Responses to RC2**

Thanks authors for considering my comments but there are still some remained major concerns that need be addressed.

 For AOD downscaling, although fine spatial resolutions, two main input DEM and LUC data have much low temporal resolutions, e.g., annual, or even many years. Thus, it is not clear what role these data play in the downscaling of daily AOD data.

Response: Thank you for your careful reading and valuable suggestions. Referring to previous research, we divide the environmental covariates into static and dynamic variables. Dynamic variables are commonly referred to as the fast change factors, and static variables are the slow change factors. The slow change factors are assumed to show no significant variation over time, and the LUCC and DEM are the typical slow change factors, especially for our study area (Arid areas and semi-arid areas are widely covered by bare land and deserts with little human disturbance, so the LUCC change is minor).

Usually, as delegates of surface properties, the LUCC is often likely to indicate the intensity of human activity and is closely related to aerosol emissions, transport, and dustfall (Fan et al., 2020; Li et al., 2022). DEM, as a delegate of terrain, with a strong correlation with surface pressure, was used to represent the dispersion condition of aerosols (Xue et al., 2021; Fan et al., 2020). In this paper, LUCC selects the median time (2010) over the study period to minimize uncertainty and DEM adopted Shuttle Radar Topography Mission 90 m Digital Elevation Model (SRTM). Through Spacefor-time substitution (Padarian et al., 2022), we combine the advantages of both

dynamic and static variables to realize AOD spatiotemporal reconstruction. Generally, static variables, similar to a baseline condition, play an initial constraint role in the downscaling of monthly AOD, while dynamic variables play a more dynamic evolution role (Yan et al., 2022). In revision, we have described static variables in more details and clarified the role played by LUCC and DEM in the process of AOD downscaling

**(Pages 9-10/Lines:213-214, 215-217, 239-241, 250-251).**

- [1] Fan, W., Qin, K., Cui, Y., Li, D., and Bilal, M.: Estimation of Hourly Ground-Level PM2.5 Concentration Based on Himawari-8 Apparent Reflectance, IEEE Transactions on Geoscience and Remote Sensing, 59, 76-85, https://doi.org/10.1109/TGRS.2020.2990791, 2020.
- [2] Li, K., Bai, K., Ma, M., Guo, J., Li, Z., Wang, G., and Chang, N.-B.: Spatially gap free analysis of aerosol type grids in China: First retrieval via satellite remote sensing and big data analytics, ISPRS Journal of Photogrammetry and Remote Sensing, 193, 45-59, https://doi.org/10.1016/j.isprsjprs.2022.09.001, 2022.
- [3] Padarian, J., Stockmann, U., Minasny, B., and McBratney, A. B.: Monitoring changes in global soil organic carbon stocks from space, Remote Sensing of Environment, 281, 113260, https://doi.org/10.1016/j.rse.2022.113260, 2022.
- [4] Xue, W., Wei, J., Zhang, J., Sun, L., Che, Y., Yuan, M., and Hu, X.: Inferring Near-Surface PM2.5 Concentrations from the VIIRS Deep Blue Aerosol Product in China: A Spatiotemporally Weighted Random Forest Model, Remote Sensing, 13, 505, https://doi.org/10.3390/rs13030505, 2021.
- [5] Yan, X., Zang, Z., Li, Z., Luo, N., Zuo, C., Jiang, Y., Li, D., Guo, Y., Zhao, W., Shi, W., and Cribb, M.: A global land aerosol fine-mode fraction dataset (2001– 2020) retrieved from MODIS using hybrid physical and deep learning approaches, Earth System Science Data, 14, 1193–1213, https://doi.org/10.5194/essd-14-1193-2022, 2022.

2. Figures S2 and S3 highlights the advantage of the generated high-resolution (250 m) AOD dataset at the city level that should be placed in the main text. But question is, why is the spatial distribution of FEC AOD opposite to MAIAC AOD? For example, FEC is low while MAIAC is high in the northwest, and southeast is just reversed (Figures S2). In addition, MxD04 with a higher spatial resolution rather than MxD08 should be used here for comparison.

Response: Thank you for your careful reading and precise advice. According to your suggestion, we have added the figures in the main text (Pages 31/Lines:630-632).

About "But question is, why is the spatial distribution of FEC AOD opposite to MAIAC AOD? For example, FEC is low while MAIAC is high in the northwest, and southeast is just reversed (Figures S2)"

Generally, FEC AOD is highly consistent with MAIAC AOD on the whole, Specially, the monthly correlations are all above 0.78 in the study area, and most of these are higher than 0.9 (N = 240, Rmean = 0.928, P

Figure S3. Monthly correlation between FEC AOD and MAIAC AOD in the study area (P

---

## Author Response (AR3)

**Comments to the author**:

Very interesting approach and outcome. Thank you for good description and for using ESSD.

I remain concerned about uncertain English, obviously not native language of authors. Copernicus publications employs highly-capable language services but this manuscript will challenge their skills. Please can authors use English-speaking colleague or one of the excellent language improvements services available in China to bring final language of this manuscript up to Copernicus standards. Not a substantial re-write, but too many small errors remain.

Response: Thank you for your recognition and advice. In revision, we turned to professional language improvement services in American Journal Experts (AJE), where editors are native English-speaking professionals. We hope the quality of current manuscript can meet the Copernicus standard.

[Figure]

**Notification to the authors**:

I've just noticed that your figure 1 contains aerials Please check whether an appropriate copyright/image credit is required and add it either in the figure itself or in the figure caption. If you are the originator, you can just inform us.

Response: Thank you for your notification. The five aerials were derived from an open-source platform (Google Earth, https://earth.google.com/). In revision, we have added aerials sources in Figure 1 caption (Page 6/Line: 153).